# Thioglycoligase derived from fungal GH3 β-xylosidase is a multi-glycoligase with broad acceptor tolerance

Manuel Nieto-Domínguez [1✉], Beatriz Fernández de Toro[2], Laura I. de Eugenio[1], Andrés G. Santana[3], Lara Bejarano-Muñoz [1], Zach Armstrong [4], Juan Antonio Méndez-Líter[1], Juan Luis Asensio[3], Alicia Prieto[1], Stephen G. Withers [4], Francisco Javier Cañada [2] & María Jesús Martínez [1✉]

The synthesis of customized glycoconjugates constitutes a major goal for biocatalysis. To this end, engineered glycosidases have received great attention and, among them, thioglycoligases have proved useful to connect carbohydrates to non-sugar acceptors. However, hitherto the scope of these biocatalysts was considered limited to strong nucleophilic acceptors. Based on the particularities of the GH3 glycosidase family active site, we hypothesized that converting a suitable member into a thioglycoligase could boost the acceptor range. Herein we show the engineering of an acidophilic fungal β-xylosidase into a thioglycoligase with broad acceptor promiscuity. The mutant enzyme displays the ability to form O-, N-, S- and Se- glycosides together with sugar esters and phosphoesters with conversion yields from moderate to high. Analyses also indicate that the p$K_a$ of the target compound was the main factor to determine its suitability as glycosylation acceptor. These results expand on the glycoconjugate portfolio attainable through biocatalysis.

[1] Biotechnology for Lignocellulosic Biomass Group, Centro de Investigaciones Biológicas Margarita Salas (CSIC), C/Ramiro de Maeztu 9, 28040 Madrid, Spain. [2] NMR and Molecular Recognition Group, Centro de Investigaciones Biológicas Margarita Salas (CSIC), C/Ramiro de Maeztu 9, 28040 Madrid, Spain. [3] Glycochemistry and Molecular recognition group, Instituto de Química Orgánica General (CSIC), C/Juan de la Cierva, 3, 28006 Madrid, Spain. [4] Department of Chemistry, Centre for High-Throughput Biology, University of British Columbia, Vancouver, Canada. ✉email: mnietdom@gmail.com; mjmartinez@cib.csic.es

Glycobiology is a research area in rapid expansion, but its development has been challenged by the difficulties to work with glycans of well-defined chemical structure. Efficient strategies for glycoside synthesis are mandatory for this purpose. Considerable efforts have been invested in the development of both chemical and biochemical approaches to afford this task, including elaborated chemoenzymatic multistep sequential synthesis or the application of automated devices for the assembly of well-defined oligo- and polysaccharides[1–3]. However, the huge diversity of the chemical space in glycosciences still demands new and versatile tools towards the synthesis of ever more varied glycoconjugates, in accordance with their central role in many biological processes. These conjugates are formed by carbohydrates covalently linked to other biological molecules, leading to glycoproteins, glycopeptides, sugar esters and phosphoesters, glycolipids and glycosides[4–8]. Very different compounds can be found among the latter category, including a large variety of natural products like polyphenols or flavonoids[9], and also others more singular, like thioglycosides[10] or selenoglycosides[11]. While classical synthetic chemistry can achieve the synthesis of a particular glycoside, it often requires multiple reaction steps and hazardous reagents[12]. On the contrary, enzymatic approaches tend to be highly selective, but specific biocatalysts must be identified for each type of reaction, such as lipases for sugar fatty acids, nucleotidyltransferases for nucleotide sugars, glycoside phosphorylases and sugar kinases for sugar phosphates, and glycosyltransferases, glycoside phosphorylases and glycoside hydrolases (GHs) for glycoproteins and other glycosylated natural products[13]. In this regard, a biotool of broad spectrum may be a significant step forward, providing a simple and environmentally friendly way to exploit the great advantages of biocatalysis. Regarding the enzymes cited above, glycosidases, and especially retaining glycosidases, are particularly interesting since engineering strategies have been developed to boost their synthetic capabilities over the canonical hydrolytic reaction against carbohydrate-polymers[14,15].

Retaining glycosidases catalyze glycosidic-bond hydrolysis through a two-step mechanism, whereby the aglycone moiety of the substrate is first protonated by the catalytic acid/base residue, thus triggering its departure, while the catalytic nucleophile attacks the anomeric center through the opposite face, rendering a covalently linked glycosyl-enzyme intermediate. The second step of the process involves the attack of an incoming water molecule to the anomeric center, which is assisted by the catalytic acid/base residue, now in its carboxylate form. At this stage, the nucleophilic residue acts as a leaving group to release a hemiacetal product that maintains the original anomeric configuration as the result of two consecutive inversions. When this second attack is performed by an acceptor other than water, a new glycoside is formed (Fig. 1a). However, the newly formed glycoside is also subjected to hydrolysis, therefore glycosylation yields are often low[12]. Engineered glycosidases have been developed in order to increase those yields and, among them, thioglycoligases are obtained by replacing the catalytic acid/base residue of retaining GHs by an inert amino acid. In the absence of their acid/base residue, thioglycoligases require both an activated glycosyl donor to form the glycosyl-enzyme intermediate and an acceptor that is a strong nucleophile, capable of reacting without basic catalytic assistance (Fig. 1a); consequently, nucleophiles successfully assayed in the literature were almost exclusively small anions, such as azide or acetate, and thiols[12,16,17]. Despite the fact that thioglycosides are interesting non-hydrolyzable sugar surrogates and they can be produced in high yields using these mutants, the requirement of strong nucleophilic acceptors caused the scope of thioglycoligases to be defined as "limited"[12]. More recently, several phenols[18] and an extended set of carboxylic acids[19] have been

tested as acceptors, expanding the scope of the reaction, which, even so, is still limited. In spite of the large repertoire of glycosidases classified in more than 160 families to date (CAZy data base), research on thioglycoligases has involved just a few enzymes from GH families 1, 2, 11, 13, 20, 31, 35, 51, and 89[20–29], most of which display maximal activity in a pH range from 6 to 8 and are unable to function under more acidic conditions (Supplementary Table 1).

In this work, we propose enzymes from CAZy family GH3 as an ideal source of thioglycoligases and analyze this hypothesis through the engineering of BxTW1, an acidophilic fungal β-xylosidase from *Talaromyces amestolkiae*. We find out that the mutant enzyme displays a broad acceptor range, forming a wide variety of glycosides of different nature. The assays reveal that the feasibility of the glycosylation reaction depends mostly on the $pK_a$ of the potential acceptor.

## Results

**Insight into the GH3 family active site**. GH3 family contains, among others, some glycosidases that are active at markedly low pH values. 3D structures of GH3 enzymes indicated the presence of an additional acid residue in the active site (Fig. 1b). This residue is conserved in at least 200 members of the family and directed mutagenesis studies revealed a measurable effect on enzyme activity[30,31], which led to suggest that it may participate in the catalytic mechanism[31]. We hypothesized that a thioglycoligase with these properties may be able to function in an acidic environment instead of the conventional neutral/basic pH. Under these conditions, many potential acceptors, such as carboxylic acids and phosphorylated derivatives would be neutral and therefore able to enter the enzyme's active site, thus avoiding any potential electrostatic repulsion which may limit the range of possible acceptors. Once there, the acceptor may transfer a proton to the secondary acid residue, becoming an enhanced nucleophile capable of attacking the glycosyl-enzyme intermediate.

**Conversion of rBxTW1 into a thioglycoligase**. The enzyme selected as model in the present study was rBxTW1, a β-xylosidase from the ascomycete *T. amestolkiae* expressed in *Pichia pastoris*[32]. This decission was based on the pH profile of the enzyme and its high acceptor promiscuity in transxylosylation reactions. On the one hand, the optimum pH of rBxTW1 was determined to be as low as pH 3, with the enzyme remaining active and stable between pH 2.2 and 6.0. In this regard, three basic residues were identified close to the catalytic acid/base and the catalytic nucleophile of rBxTW1 (Fig. 1c). These basic amino acids may decrease the $pK_a$ of the catalytic residues by stabilizing their anionic states through the formation of salt bridges[33]. Such interactions might explain the low optimum pH of rBxTW1. On the other hand, the range of acceptors tested with the wild-type rBxTW1 comprised a wide variety of compounds, from simple alkanols to bulky hydroxynaphthalenes[32,34]. The transferred monosaccharide, xylose, was also relevant for the choice of rBxTW1. Xylose is the most abundant pentose in nature and a central sugar in the plant kingdom. There are many xylosyl phenolic derivatives involved in plants' metabolism but their role is poorly understood since both their chemical synthesis and their purification from plant tissues are difficult and low yielding[35,36]. Therefore, enzymatically obtained β-xylosides can facilitate research on plant metabolites and, from a broader perspective, complement the available collection of biocatalytically synthesized glyconjugates, the vast majority of which are generated by the attachment of hexoses[37,38]. In addition, xylose also possesses a technical advantage since, due to the absence of ramifications in pentopyranoses, *p*NPX is intrinsically more reactive than its

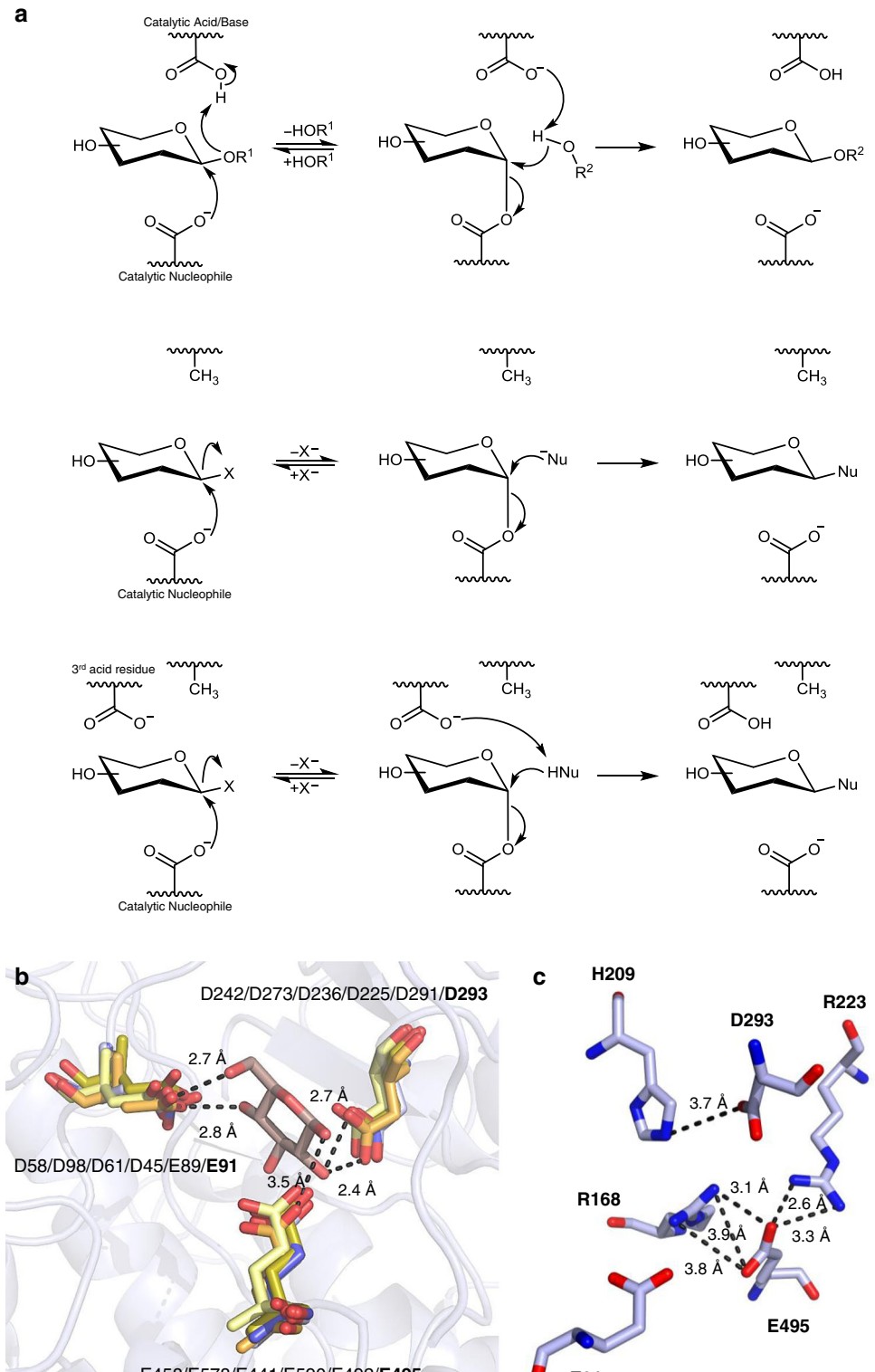

hexose derived counterparts, which circumvents the use of dinitrophenyl sugar derivatives that are much more unstable and difficult to handle[39].

The primary acid/base catalytic residue of this β-xylosidase (E495) was replaced by Ala, Gly, and Gln to obtain thioglycoligase mutants. A typical characterization of the biocatalysts was carried out. Based on kinetics (Supplementary Table 2) and on the effect of small anions on the activity (Supplementary Fig. 1) the rBxTW1-E495A mutant was selected as the most promising

biocatalyst. As commented in the Supplementary Discussion, wide differences on kinetic performance were observed depending on the selected buffers at the same pH. These findings were the first indication that the buffering agent itself may participate in the reaction, a phenomenon closely intertwined with the proposed mode of action of the mutant, which will be further discussed below. Additionally, the ability of this mutant to synthetize thioglycosides as a conventional thioglycoligase was confirmed using thiophenol (Supplementary Fig. 2). Once

**Fig. 1 Mechanistic and structural foundations of the proposed hypothesis.** Mechanism of retaining β-glycosidases (**a**), structural insights on GH3 enzymes (**b**), and view of the active site of a rBxTW1 model (**c**). **a** From up to down, glycosylation and deglycosylation catalyzed by wild-type enzyme; typical thioglycoligase; and GH3 thioglycoligase, where the third acid residue of the latter family is proposed to participate in the reaction mechanism somewhat facilitating the neutralization of the leaving group or accepting a proton from the potential glycosyl acceptor. Nu: nucleophile. **b** Overlap of five solved structures from the GH3 family and modeled rBxTW1 showing the three conserved acid residues of the active site and a β-D-glucose ligand. E: Catalytic acid/base (Glu); D: Catalytic nucleophile (Asp); E/D: Conserved acid residue (Glu for β-xylosidases and Asp for β-glucosidases). Residue positions are displayed according to the following order: β-glucosidase pdb:2X40[31] (pale yellow); β-glucosidase pdb:4I3G[82] (bright orange); β-glucosidase pdb:3ZYZ[83] (sand); and β-glucosidase pdb:3AC0[84] (olive) β-xylosidase pdb:5A7M (blue) and model (SWISS-MODEL tool) of β-xylosidase rBxTW1 (light blue). A glucose ligand and its polar contacts with the corresponding β-glucosidase pdb:3ZYZ are displayed[83]. The background shows the residues' environment in rBxTW1. The solved structures are referred by their PBD IDs. **c** Acid and basic residues in the active site of the rBxTW1 model. The catalytic nucleophile, the catalytic acid/base and the third acid residue are identified as D293, E495, and E91 respectively. The distances between the atoms potentially involved in salt bridges are indicated. Generation of protein images and distance measurements were performed using PyMOL Molecular Graphics System version 1.3.

established the functionality of the thioglycoligase, the work was structured to first determine its acceptor tolerance, then explain the results under the light of the proposed hypothesis for GH3 thioglycoligases, and finally perform a quantitative assessment of the glycosylation yields.

**Acceptor range of rBxTW1-E495A.** Following this schedule, the purified protein was assayed against a library of potential acceptors selected on the basis of a wide range of $pK_a$ values, comprising a variety of functional groups, and readily available in the laboratory. Tables 1 and 2 display the results of the assessment of glycoside formation by thin layer chromatography (TLC) and mass spectrometry (ESI-MS), and the maximum conversion yield for those acceptors which were later quantitatively studied. Some representative examples of the whole diversity of functional groups assayed were selected to be purified and identified by NMR and confirmed through their molecular masses (Fig. 2, Supplementary Figs. 3–7, and Supplementary Methods).

An overall analysis of the table reveals the great degree of promiscuity of the mutant enzyme. The list of obtained compounds includes O-, N-, S-, and Se- glycosides together with sugar esters and phosphoesters. O-Glycosides comprised not only C–O–C but also C-O-N glycosidic linkages.

This diversity goes far beyond the range of the wild-type enzyme. It may be associated to the fact that mutating the catalytic acid/base residue widens the subsite +1, which might loosen the constraints for acceptor accommodation[24]. In addition, the model of rBxTW1 also indicates that subsite +1 is considerably shallow, being delimited by a few residues: Q14, S15, Q16, P17, L24, and R98 (Supplementary Fig. 8), which may facilitate the approach of the acceptor to the active site. According to our hypothesis, the potential ease of acceptor entrance caused by the mutation is combined with the effect of the third acid residue, which could mimic the function of the missing catalytic acid/base allowing the disruption of the glycosyl-enzyme intermediate without requiring an acceptor with strong nucleophilicity.

**Role of $pK_a$ on the acceptor suitability.** Surprisingly, a broad analysis of the results from Tables 1 and 2 indicates that the influence of $pK_a$ of the acceptor is much more decisive than its particular nature or structure. The enzyme seems to be able to catalyze the glycosylation of any functional group with a $pK_a$ value between 2 and 9. As it can be inferred from the table, the range refers to functional groups neutral in their protonated state. Considering the proposed hypothesis, ionization constants above $pK_a$ 10 may prevent the deprotonation of the acceptor at the active site in the pH range of 2.2 to 5.5, in which rBxTW1 is stable and highly active[32]. Those acceptors would require at least a reaction pH ~7–8 to allow their deprotonation, but rBxTW1 is

not stable at those pH values, and residual activity is very low even at short reaction times[32]. Consequently, both due to the enzyme instability and a probable catalytic impairment of the first step of the double displacement mechanism, the glycosyl-enzyme intermediate would not be formed at those pH values. Potential acceptors with $pK_a$ values below 1 did not render glycosylation producs either because their anionic nature causes electrostatic repulsion in the active site[40], or because they are inherently poor nucleophiles, or because any product formed would likely hydrolyze spontaneously under the employed reaction conditions. For acceptors with $pK_a$ values in the range of 9–10 or 1–2, little or no visible spots were detected by TLC, although the expected xylosides were detected by ESI-MS suggesting that they are synthetized in small amounts. Careful analysis of these data revealed a few exceptions: formic acid, benzeneseleninic acid, N-hydroxysuccinimide, 2-(N-morpholino)ethanesulfonic acid (MES), and L-glycine display $pK_a$ values inside the suitable range for the mutant, but no glycosylated products were detected by TLC or ESI-MS analyses. In the case of L-glycine and MES, their zwitterionic nature must be taken into account. For these compounds, rather than product instability, the proximity of a formal charge to the potential accepting group under the assay conditions may simply prevent their binding to the active site in a productive pose to act as acceptors. This latter rationale correlated with the fact that increasing MES concentrations were unable to boost the activity of the mutant enzyme (Supplementary Fig. 9). Similarly, L-glutamic acid gave a single xyloside, whereas N-acetyl-L-glutamic acid, without an ionizable amino group, produced two xylosides, compound 2 (Fig. 2) and another product too unstable to be characterized by NMR, but most probably corresponding to the sugar ester at the α carboxyl group. These findings were exploited by using glycine-HCl and MES for buffering the samples and to control the reaction pH without interfering with the activity of the mutant enzyme.

Conversely, the absence of new xylosides from formic acid, benzeneseleninic acid, and N-hydroxysuccinimide, compounds also within the $pK_a$ range from 2 to 9, could be explained merely by product instability. Thus, the glycoside might form, but is probably getting rapidly degraded, which would prevent its accumulation and detection[17]. To test this hypothesis, a series of ligase activity response assays were set up for both the mutant and the wild-type enzyme, using increasing concentrations of the 12 selected acceptors (Fig. 3). This assay is based on the fact that the disruption of the glycosyl-enzyme intermediate is the rate-limiting step for rBxTW1-E495A using pNPX as donor[14]. An acceptor able to speed up this disruption regenerates the ability of the enzyme to catalyze the cleavage of a new molecule of pNPX, producing an activity enhancement that can be measured through the release of pNP under steady-state conditions (Supplementary Fig. 10). The assays included the wild-type rBxTW1 as a control

**Table 1 Screening of acceptors for rBxTW1-E495A: carboxylic and alcohol target functional groups.**

| Functional group | Acceptor | p$K_a$ | TLC | ESI-MS | Yield (%) |
|---|---|---|---|---|---|
| R–COOH | L-Glutamic acid | 2.10; 4.07; 9.47 | ✓ | ✓ | |
| | L-Glycine | 2.35; 9.78 | ✗ | ✗ | |
| | N-Acetyl-L-glutamic acid (1) | 3.45[a]; 4.29[b] | ✓ | ✓ | |
| | Formic acid | 3.75 | ✗ | ✗ | |
| | 5-Aminolevulinic acid | 3.90; 8.05 | ✓ | ✓ | |
| | trans-Cinnamic acid (3) | 4.44 | ✓ | ✓ | 33.3 |
| | Acetic acid | 4.76 | ✓ | ✓ | |
| | Propionic acid | 4.87 | ✓ | ✓ | |
| | Butyric acid | 4.82 | ✓ | ✓ | |
| | Nicotinic acid | 4.82 | ✓ | ✓ | |
| | Pentanoic acid | 4.86 | ✓ | ✓ | |
| | Hexanoic acid (5) | 4.88 | ✓ | ✓ | |
| | Heptanoic acid (7) | 4.89 | ✓ | ✓ | |
| | Octanoic acid | 4.95 | ✓ | ✓ | |
| | Cholic acid | 5.08 | ✓ | ✓ | |
| R–COOH / R–OH | Gallic acid | 4.24; 8.27; 9.23 | ✓ | ✓ | |
| | p-Coumaric acid (38) | 4.39; 8.37 | ✓ | ✓ | 94.6 |
| | Sinapic acid | 4.40; 9.21 | ✓ | ✓ | |
| | Caffeic acid | 4.47; 8.32 | ✓ | ✓ | |
| | Ferulic acid (34) | 4.56; 8.65 | ✓ | ✓ | 100.0 |
| R–OH | Quercetin | 5.54; 6.95; 8.21; 9.77 | ✓ | ✓ | |
| | Silibinin | 7.00; 8.77; 9.57; 11.66 | ✓ | ✓ | |
| | Vanillin (9) | 7.40 | ✓ | ✓ | 65.5 |
| | Phloretin | 7.58; 9.29[b]; 9.89[b]; 10.99[b] | ✓ | ✓ | |
| | Epigallocatechin gallate (EGCG) (12) | 7.75; 8.0 | ✓ | ✓ | 30.8 |
| | Methyl 4-hydroxybenzoate | 8.31[a] | ✓ | ✓ | |
| | Pyrogallol | 9.01; 11.64 | ✓ | ✓ | |
| | 2,6-Dihydroxynaphthalene | 9.55[a] | ✓ | ✓ | |
| | 2,6-Dimethoxyphenol | 9.95 | ✗ | ✗ | |
| | Pterostilbene | 9.96[a] | ✓ | ✓ | |
| | Phenol | 9.99 | ✗ | ✓ | |
| | D-Xylose | 12.29 | ✗ | ✗ | |
| | D-Glucose | 12.46 | ✗ | ✗ | |

✗: No product detected, ✓: Product detected. Bold numbers are referred to Fig. 2 and indicate that the products from those acceptors were additionally identified by NMR.
[a] Predicted values from SciFinder.
[b] Predicted values from ChemAxon.

to detect any other effect of the potential acceptors on enzymatic activity, since no significant acceleration of the catalysis is expected when the functional acid/base residue is present. Compounds outside the p$K_a$ range from 2 to 9 should not influence the specific activity of the mutant biocatalyst, whereas acceptors with appropriate ionization constants were expected to increase its activity, regardless of whether the corresponding product is detected or not. Indeed, the activity profiles displayed in Fig. 3 completely agree with these expectations. No activity increase was perceived for ethanesulfonic acid (p$K_a = -1.68$), acetophenone oxime (p$K_a = 11.48$), and D-glucose (p$K_a = 12.46$), and slight activity enhancements were noted for phenylphosphonic acid (p$K_a = 1.42$) and 1H-1,2,3-triazole (p$K_a = 9.26$). However, a concentration-dependent activity response was observed for benzenesulfinic acid (p$K_a = 2.76$), hydrazoic acid (p$K_a = 4.72$), vanillin (p$K_a = 7.40$), 1H-1,2,3-triazolo[4,5-b]pyridine (p$K_a = 8.70$), formic acid (p$K_a = 3.75$), benzeneseleninic acid (p$K_a = 4.70$), and N-hydroxysuccinimide (p$K_a = 6.09$), even though xylosides of the three latter compounds were not detected. These results indicate that the three potential acceptors behave as expected for compounds within the p$K_a$ range from 2 to 9 and that the most plausible reason for not detecting the corresponding xylosyl derivatives is indeed their instability. The only ligand that increased the activity of the wild-type enzyme was glucose, presumably via standard transxylosylation reactions[34].

**Relation between acceptor p$K_a$, reaction pH, and glycosylation.** Once established the acceptor scope of this biocatalyst, the

**Table 2 Screening of acceptors for rBxTW1-E495A: N-, P-, S-, and Se-containing target functional groups.**

| Functional group | Acceptor | $pK_a$ | TLC | ESI-MS | Yield (%) |
|---|---|---|---|---|---|
| R$^1$—N—OH, R$^2$ | 6-Chloro-1-hydroxybenzotriazole | 4.15 | ✓ | ✓ | |
| | Violuric acid | 4.57 | ✓ | ✓ | |
| | N-Hydroxybenzotriazole (14) | 4.60 | ✓ | ✓ | 100.0 |
| | N-Hydroxysuccinimide | 6.09 | ✗ | ✗ | |
| | N-Hydroxyphthalimide | 6.32 | ✓ | ✓ | |
| | Acetophenone oxime | 11.48 | ✗ | ✗ | |
| R$^1$—NH, R$^2$ | 5-Trifluoromethyl-2H-tetrazole | 2.27[a] | ✓ | ✓ | |
| | 5-(5-Bromo-2-thienyl)-1H-tetrazole (16) | 3.57[a] | ✓ | ✓ | 97.7 |
| | Hydrazoic acid (18) | 4.72 | ✓ | ✓ | |
| | 4-Phenylurazole | 5.29 | ✓ | ✓ | |
| | 3,5-Dibromo-1,2,4-triazole (20) | 5.50[a] | ✓ | ✓ | 100.0 |
| | 3-Nitro-1H-1,2,4-triazole | 5.92[a] | ✓ | ✓ | |
| | 5-Chlorobenzotriazole | 7.7 | ✓ | ✓ | |
| | 1,2,4-Triazolo[4,3-a]pyridin-3(2H)-one | 7.86[a] | ✓ | ✓ | 98.8 |
| | 1H-Benzotriazole | 8.2 | ✓ | ✓ | |
| | 1H-1,2,3-Triazolo[4,5-b]pyridine (22) | 8.70[b] | ✓ | ✓ | |
| | 1H-1,2,3-Triazole | 9.26 | ✓ | ✓ | |
| | 1H-1,2,4-Triazole | 10.04 | ✗ | ✓ | |
| R—P(=O)(OH)OH | Phenylphosphonic acid (24) | 1.42; 6.92 | ✓ | ✓ | |
| | Phosphoric acid (26) | 2.12; 7.21; 12.32 | ✓ | ✓ | 31.5 |
| R—SH | Thiophenol (28) | 6.62 | ✓ | ✓ | 99.9 |
| R—S(=O)OH | Benzenesulfinic acid (30) | 2.76 | ✓ | ✓ | 71.2 |
| R—S(=O)(=O)OH | Methanesulfonic acid | -1.92 | ✗ | ✗ | |
| | Ethanesulfonic acid | -1.68 | ✗ | ✗ | |
| | 2-(N-morpholino)ethanesulfonic acid | -1.52[b]; 6.15 | ✗ | ✗ | |
| R—SeH | Benzeneselenol (32) | 4.73[a] | ✓ | ✓ | 23.1 |
| R—Se(=O)OH | Benzeneseleninic acid | 4.70 | ✗ | ✗ | |

✗: No product detected, ✓: Product detected. Bold numbers are referred to Fig. 2 and indicate that the products from those acceptors were additionally identified by NMR.
[a] Predicted values from SciFinder.
[b] Predicted values from ChemAxon.

optimal reaction conditions for glycosylation were sought out to gain more insight in the mode of action of the mutant. The analysis was performed by varying the initial substrate concentrations, pH values, and reaction times, as shown in Fig. 4. Interestingly, the influence of the pH largely fitted our hypothesis: values above the $pK_a$ of the acceptor substrate were suboptimal even though the enzyme was within its range of activity[32] and conjugated bases are stronger nucleophiles than their parent neutral acids. Thus, the highest conversion yield for benzenesulfinic acid and 5-(5-bromo-2-thienyl)-1H-tetrazole was achieved at a pH value below the acceptor $pK_a$, despite the tetrazole was only partially soluble at pH 3 and fully soluble at pH 5.5. Similar results were obtained with 5-trifluoromethyl-2H-tetrazole, a compound completely soluble in all the assayed conditions, for which ligase activity response was faster and at lower acceptor concentrations at pH 2.2 and 3.0 than at pH 5.5. The outcome was quite different for the acceptors with higher $pK_a$ values, vanillin and 1H-1,2,3-triazolo[4,5-b]pyridine, for which

lower pH values were not favorable. This may seem paradoxical and it will be discussed below, since the lower the pH value is, the more displaced the assayed acceptors are to their neutral forms.

It should be considered that the dependence of enzymatic reactions on pH, when both enzyme and ligand can be ionized, is notoriously hard to dissect. However, the results consistently support the rationale that there is a preference for the neutral form of the acceptor to enter the active site. Moreover, the data obtained for vanillin and 1H-1,2,3-triazolo[4,5-b]pyridine indicated that, besides favouring the neutral form of the acceptor, the pH value must be compatible with the deprotonation of the selected compound in the active site. Such deprotonation is probably unfavorable for vanillin and 1H-1,2,3-triazolo[4,5-b]pyridine at pH 2.2–3 because of the large difference between its dissociation constant and the reaction pH, explaining why higher performance is attained at pH 5.5, a value that still keeps the equilibrium displaced to the neutral form. Thus, the relation between $pK_a$ and pH has proven to be the key factor for the

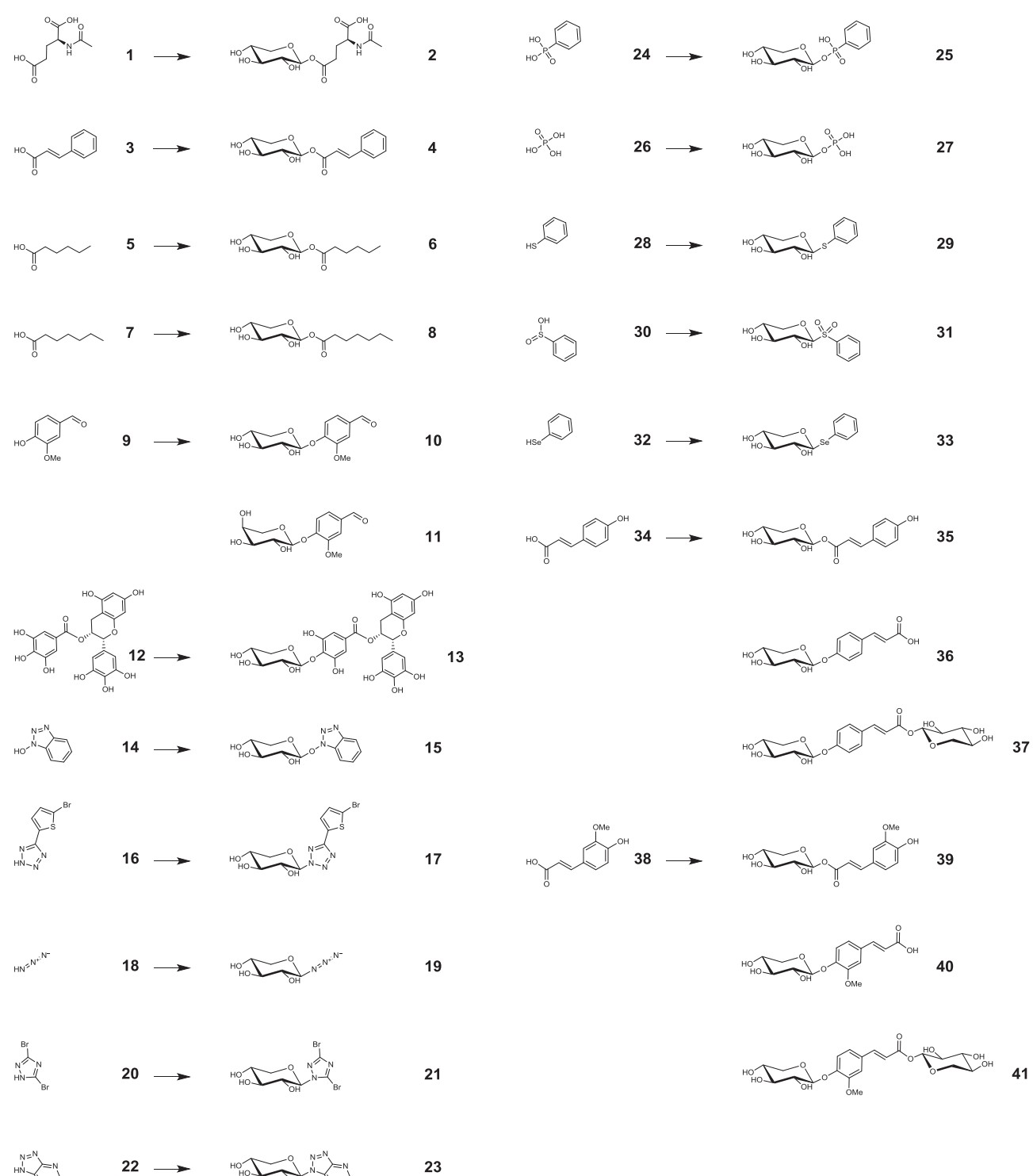

**Fig. 2 Acceptors and their derived glycosides identified by NMR.** Bold numbers are used to identify the acceptors (Tables 1 and 2) and their corresponding glycosides.

activity of rBxTW1-E495A. In summary, the pH value of a successful glycosylation reaction must satisfy the following requirements: (i) be in the range of enzyme activity and stability; (ii) keep the acceptor in an equilibrium displaced to its neutral form; (iii) allow the deprotonation of the acceptor during the nucleophilic attack.

**Glycosylation yields and potential applications.** The final question to be addressed was whether the synthetic capabilities of rBxTW1-E495A might be exploited as a truly useful biotool. To unravel the biotechnological potential of the enzyme, at least one representative acceptor from each functional group was selected among those included in Tables 1 and 2, determining the trans-xylosylation yield as a function of reaction time (Fig. 5). The assays compared the reaction conditions in a suitable buffer to the absence of pH control, by simply adjusting the initial pH to a value below the p$K_a$ of the acceptor, to assess the glycosylation yield in a non-buffered environment.

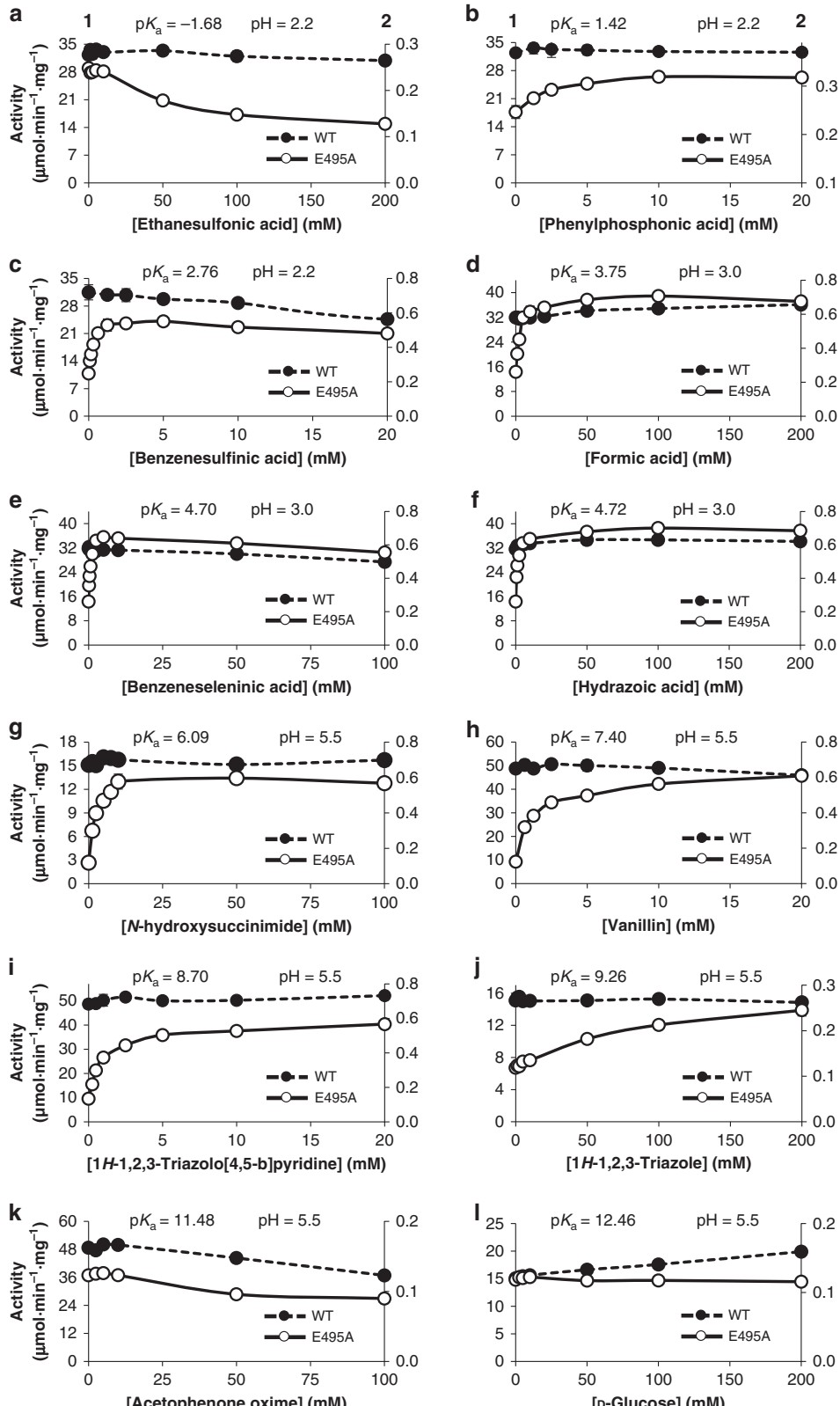

**Fig. 3 Ligase activity response assays.** Specific enzyme activity was evaluated in the presence of increasing concentrations of some potential acceptors. Selected compounds were ethanesulfonic acid (**a**), phenylphosphonic acid (**b**), benzenesulfinic acid (**c**), formic acid (**d**), benzeneseleninic acid (**e**), hydrazoic acid (**f**), N-hydroxysuccinimide (**g**), vanillin (**h**), 1H-1,2,3-triazolo[4,5-b]pyridine (**i**), 1H-1,2,3-triazole (**j**), acetophenone oxime (**k**), and D-glucose (**l**). p$K_a$ of each compound and buffered reaction pH values are indicated. Activities were calculated spectrophotometrically against pNPX and expressed as μmol of released pNP per min and mg of enzyme. Values for wild-type enzyme (WT) are referred to y-axis-1 while y-axis-2 corresponds to rBxTW1-E495A. Activity profiles are represented by a dashed line and black circles for the wild-type enzyme and by a continuous line and white circles for rBxTW1-E495A. Mean values are shown together with the corresponding standard error (n = 2 independent experiments). Source data are provided as a Source Data file.

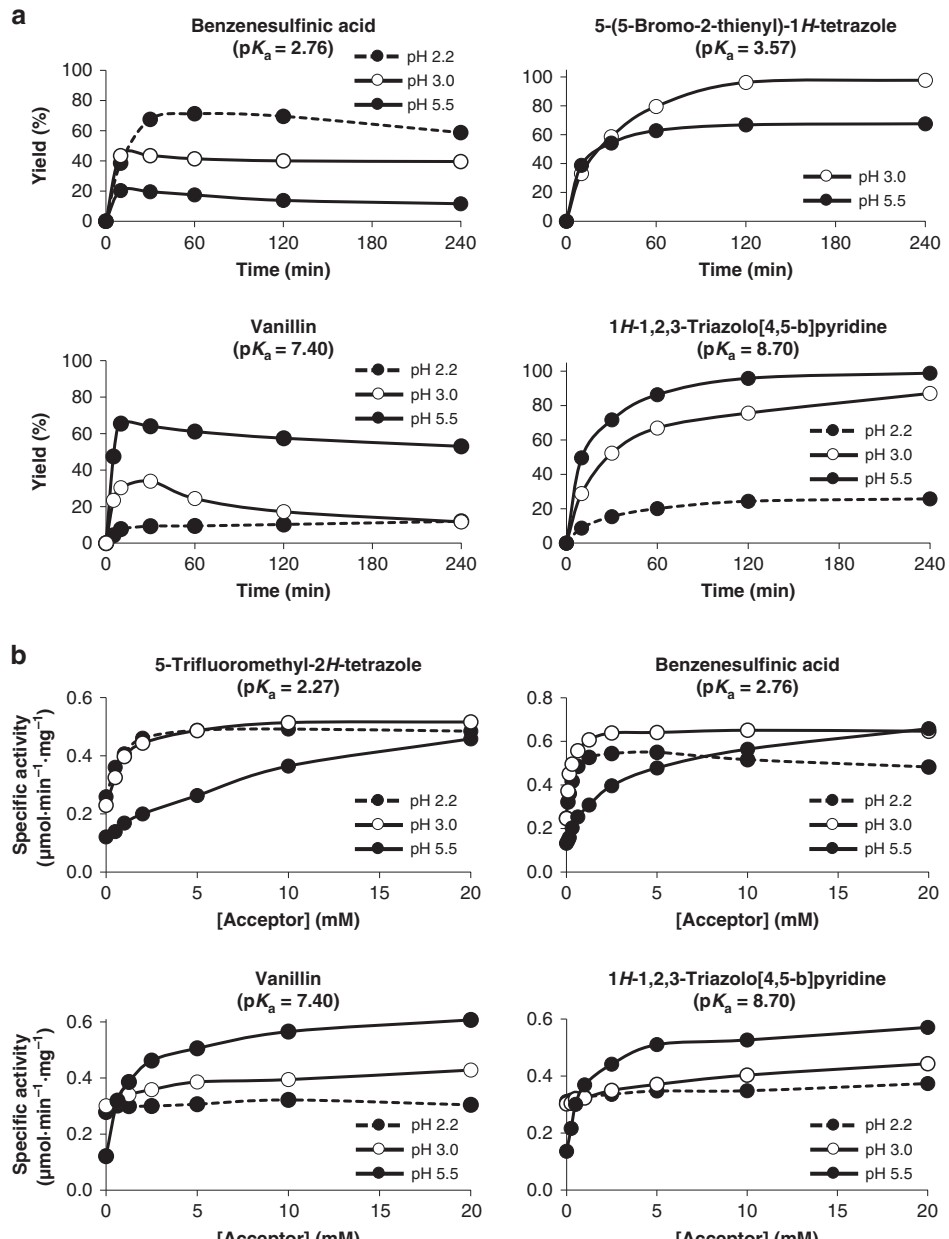

**Fig. 4 Relationship between p$K_a$ and pH with the activity of rBxTW1-E495A. a** Yield of acceptor conversion (%) into a xylosyl derivative in the presence of donor excess. The initial donor/acceptor ratio was 2/1 (mole/mole) and the acceptor conversion was determined by HPLC at different time intervals for 4 h. **b** Ligase activity response assay for each acceptor. The effect of each acceptor on the activity of rBxTW1-E495A was assayed over a range of acceptor concentrations. β-Xylosidase activity was determined spectrophotometrically using pNPX as substrate. Determined profiles are represented by a dashed line and black circles for pH 2.2, by a continuous line and white circles for pH 3.0 and by a continuous line and black circles for pH 5.5. Mean values are shown together with the corresponding standard error ($n = 3$ independent experiments). Source data are provided as a Source Data file.

The results revealed an unquestionable biotechnological potential, with most production values in the range from moderate to high, including three examples attaining full conversion. From a synthetic perspective, the capacity of the mutant to selectively attach a xylose moiety to a flavonoid such as EGCG is probably the most interesting trait. Flavonoids are polyphenols acting as secondary metabolites in plant metabolism[41] and associated to a plethora of benefits for human health[42]. Despite the fact that the effect of attaching a xylose moiety (or any other carbohydrate) on the bioactivity of the aglycon is hardly predictable, it is worth mentioning that the enzymatic glycosylation of EGCG was attained with high selectivity (compound 13), and its derived glycoconjugates have been reported to have

interesting biological activites[43]. Other proven acceptors for rBxTW1-E495A, silibinin, and quercetin, are also flavonoids with known derived bioactive glycosides[9,44]. More interestingly, there are many other polyphenols with recognized biological activities fulfilling the p$K_a$ requirements of the enzyme, which are therefore expected to be glycosylated by rBxTW1-E495A[45–49].

In addition to xylosides of flavonoids, the potential outcomes of the remaining xylo-conjugates can be summarized based on similar glycosides described in other studies. Thus, enzymatically synthesized sugar fatty acid esters are considered a green alternative to petroleum-derived surfactants. Among other properties, they have been reported to be biodegradable, non-toxic, and antimicrobial compounds[4]. Several examples of these

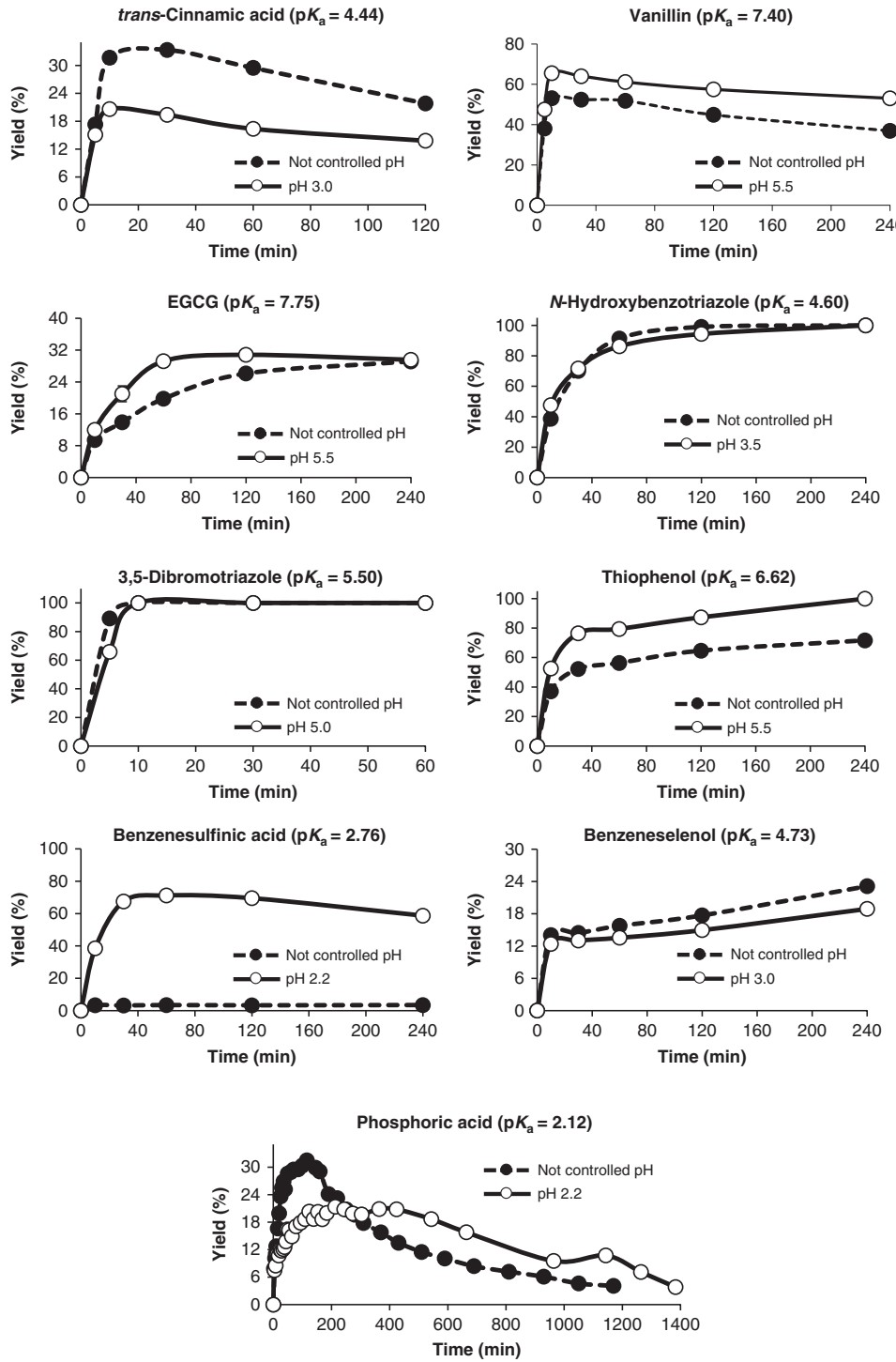

**Fig. 5 Production of different xylosides with and without pH control.** The plots present the acceptor conversion (%) over time, determined by HPLC, using a 2/1 (mole/mole) donor/acceptor balance for every acceptor, except for phosphoric acid. The latter was analyzed by NMR and yield was calculated in terms of donor conversion (%) using a 1/2 (mole/mole) donor/acceptor balance. Yield profiles are represented by a dashed line and black circles for non-controlled pH and by a continuous line and white circles for controlled pH. Mean values are shown together with the corresponding standard error ($n = 3$ independent experiments) for every acceptor except phosphoric acid. Formation of the xylosyl phosphoester was assayed with a single replicate by continuously measuring the corresponding sample over time. However, due to the testing of multiple timepoints and two different conditions, results are still significant to verify the formation and subsequent consumption of the product. Source data are provided as a Source Data file.

glycoconjugates have been described, including some derived from short- or medium-chain length carboxylic acids[4,50,51] similar to the ones successfully tested as acceptors for rBxTW1-E495A (Table 1), and even a xylosyl hexanoate ester[52] equivalent to compound 6 obtained in the present work.

Little is known about the applications of glycosyl derivatives of triazoles and tetrazoles, although almost countless different glycoconjugates may be enzymatically formed due to the huge diversity of these aglycons. Interestingly, several studies on the synthesis of a number of these derivatives have been published,

frequently assaying them for different bioactivities of interest[53–57]. Among these activities, the antimicrobial effect reported using triazole derivatives of arabinose[57] is particularly interesting, since this carbohydrate is a pentose similar to xylose.

Thioglycosides, selenoglycosides, and glycosyl sulfones act as glycosidase inhibitors, and this property confers them different bioactivities of pharmacological relevance[58–61]. This is also applicable to glycosides derived from aromatic acceptors[62,63], such as compounds 29 and 31 synthesized in this work. It should be noticed that there is a reported example of a sulfone glycoside, whose carbohydrate moiety is again arabinose[59].

The potential applications of O-xylosyl phosphates or O-xylosyl phosphonates are less clear. Despite the fact that sugar 1-phosphates are products with excellent biotechnological potential, most of the studied compounds are α-sugars and when it comes to the β-anomers the available information on biological roles is mainly restricted to β-glucose 1-phosphate[64]. However, two wild-type retaining β-glycosidases were recently reported to act as true glycoside phosphorylases. Both enzymes successfully catalyzed the synthesis of not only β-glucose 1-phosphate, but also β-N-acetylglucosamine 1-phosphate[65] or β-N-acetylglucosaminide 1-phosphate[66], suggesting that other 1-phosphorylated β-sugars might also have a biological function yet to be determined[65,66].

The analysis of glycosylation yields indicated that the highest conversions were attained for those acceptors whose corresponding xylosides were not susceptible of hydrolysis by the mutant (C–N, C–O–N, and C–S linkages). Moreover, the results confirmed the expected differences between buffering the reaction or not. As predicted, due to the huge influence of pH in the glycosylation mechanism, the yields were generally higher for buffered reactions than in their non-buffered counterparts. In the latter case, the continuous release of $p$NP, with higher p$K_a$, was expected to cause a progressive alkanization that turns the acceptor into its anion, hindering glycosylation. This is corroborated by the fact that bezenesulfinic acid, a compound with both low p$K_a$ and low self-buffering capacity, displayed a negligible glycosylation yield in the absence of buffer. However, interestingly, the non-buffered reactions with $trans$-cinnamic and phosphoric acid gave the highest xylosylation yield. Both acceptors are expected to form glycoconjugates with good leaving groups and, as a consequence, the glycosylation/hydrolysis balance makes more difficult to estimate the most suitable pH.

The analysis of glycosylation yield was extended to $p$-coumaric acid and ferulic acid. These abundant hydroxycinnamic acids have a carboxyl and a hydroxyl group, making them complex acceptors, so their glycosylation yields were studied using a central composite design[67]. Beyond their structural interest, both the aglycons and their derivatives have multiple pharmacological properties[68,69] and, therefore, were considered acceptors of biotechnological relevance. The optimum yields for both compounds were ~100%, summing up the amounts of the three products observed for each acceptor, as displayed in Fig. 2 (35, 36, and 37 for coumaric; 39, 40, and 41 for ferulic). Short reaction times and low initial pH favored the formation of the sugar ester, while increased reaction times and higher initial pH values supported the glycosylation at the aromatic hydroxyl position (Supplementary Table 3 and Supplementary Discussion). Inadvertently, these results reinforced the relationship between reaction pH and the desired accepting functional group p$K_a$ proposed above, although the potential degradation of the esters at basic pH was also a possible explanation. To analyze this alternative, the stability of the xylose ester of $trans$-cinnamic acid (from which ferulic and $p$-coumaric acid are derived) was studied at pH 7.0 and 8.0 (Supplementary Fig. 11), demonstrating that physicochemical instability, although existing, cannot explain the

wide differences in yield observed between reactions conducted at acidic and basic pH. In this regard, it should be noticed that, apart from its broad acceptor promiscuity, the preferred acidic pH range for rBxTW1-E495A from 2.2 to 5.5 has the additional advantage of enhancing the stability of both the donor and some products. Namely, xylosyl esters and xylosyl phosphoesters are sensitive to neutral pH, therefore even in the case that one of the few known examples of thioglycoligases forming C–O–C linkages[23,70] was able to use a phosphoric or a carboxylic acid acceptor, the production of the corresponding sugar ester would be compromised by their neutral or alkaline reaction pH (Supplementary Table 1).

**Alternative sugar donors**. Furthermore, the ability of rBxTW1 to employ donors other than $p$NPX was also evaluated. 4-Nitrophenyl α-L-arabinofuranoside and 4-nitrophenyl α-L-arabinopyranoside were selected to replace $p$NPX, since they behaved as secondary substrates of the wild-type enzyme[32]. Vanillin was selected as potential acceptor based on the reported bioactivities of its glyconjugates[71], together with its structural simplicity, which was expected to allow a straightforward determination of both arabinosylation and disruption of the arabinosyl derivatives. The results demonstrated the synthesis of vanillyl arabinopyranoside and arabinofuranoside, although yields were lower than those attained for vanillyl xylopyranoside. The arabinofuranoside derivative was barely detected by TLC after 4 h; instead, after 16 h the product was better visualized and the expected molar mass was confirmed by ESI-MS. The low yield qualitatively observed is likely associated with the little consumption of $p$NPAf (Supplementary Fig. 12).

The synthesis of the vanillin arabinopyranoside was confirmed by TLC both after 4 and 16 h of reaction, and further corroborated by ESI-MS. Since it was produced in higher yields than the arabinofuranoside, the product was purified and identified by NMR (compound 11) and the reaction profile was analyzed by HPLC (Fig. 6).

The production profile proved that, even after increasing the incubation time, the maximal conversion yield (47%) was lower than the one attained for the xylopyranoside (66%), probably due to the arabinopyranoside derivative being a poor substrate in comparison with its xyloside counterpart. Kinetic analysis supported this hypothesis indicating that rBxTW1-E495A displays $K_m$ and $V_{max}$ values against $p$NPAp of 5.0 ± 0.4 mM and 0.029 ± 0.001 µmol min$^{-1}$ mg$^{-1}$ respectively. Therefore, the catalytic efficiency was determined to be $1.91 × 10^{-2}$ mM$^{-1}$ s$^{-1}$, a value ~200-fold lower in comparison to the one obtained for $p$NPX under the same conditions.

## Discussion

The synthetic prowess of rBxTW1-E495A may be comparable to the high yields, regio- and stereospecificity reported for actual glycosyltransferases[7]. These enzymes have traditionally sparked attention for being the biocatalysts responsible for synthesizing most of the natural glyconjugates. In the last decade, efforts have been made to increase their acceptor range, successfully reporting glycosyltransferases able to form O-, C-, S-, and/or N-glycosidic bonds[72–75]. However, the application of glycosyltransferases is still hampered by the high cost of the activated sugars required as glycosylation donors. On the contrary, nitrophenyl-sugars are more cost-effective donors, easier to handle, and synthesize[76], which implies an advantage of rBxTW1-E495A over glycosyltransferases as a versatile synthetic biotool. The development of strategies of enzyme immobilization and recycling will probably be the next steps to improve the cost-effectiveness of the developed biotool.

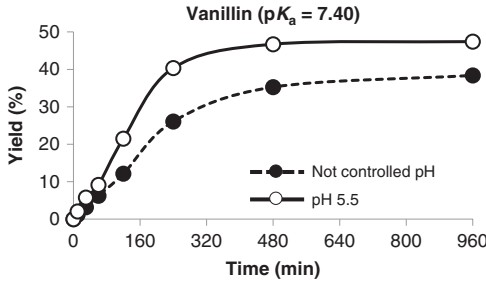

**Fig. 6 Production of vanillin arabinopyranoside with and without pH control.** The initial donor/acceptor balance was 2/1 (mole/mole) and the acceptor conversion (%) over time was determined by HPLC at different time intervals for 16 h. Yield profiles are represented by a dashed line and black circles for non-controlled pH and by a continuous line and white circles for controlled pH. Mean values are shown together with the corresponding standard error ($n = 3$ independent experiments). Source data are provided as a Source Data file.

Based on the former analyses, the main limitation to the use of rBxTW1-E495A seems to be the preferential selectivity for xyloside donors. However, it is likely that other members of the GH3 family with similar properties to the wild-type biocatalyst could be converted into thioglycoligases that exhibit an analogous behavior. For instance, a possible candidate for mutagenesis and conversion into a thioglycoligase is the β-glucosidase BGL-2 from the same fungus, *T. amestolkiae*. This enzyme shares the acidic pH profile of rBxTW1[77], which may allow obtaining a useful biotool for the transfer of glucopyranose units. Another alternative may be to further evolve rBxTW1-E495A itself to expand its donor versatility, an approach successfully applied to other glycosidases[78,79].

The mutant enzyme reported in this work has demonstrated a broad acceptor promiscuity, as well as moderate to high production yields. This study has collected the synthesis of more than 50 different glycosides using a single biocatalyst, including an arabinopyranosyl derivative. However, the chemical space of prospective acceptors ranging from p$K_a$ 2 to 9 is almost unlimited so the potential of this biotool, or an equivalent mutant obtained from another glycosidase of the same GH family, has barely been explored. The studies performed with rBxTW1-E495A may be considered as a proof of concept for the generation of multi-glycoligases obtained from the acidophilic members of the GH3 family.

## Methods

**Materials**. *N*-hydroxysuccinimide, *N*-hydroxyphtalimide, *N*-hydroxybenzotriazole, 6-chloro-1-hydroxybenzotriazole, 1*H*-1,2,3-triazolo[4,5-*b*]pyridine, 4-phenylurazole, 3,5-dibromo-1,2,4-triazole, 3-nitro-1*H*-1,2,4-triazole, 5-chlorobenzotriazole, 1,2,4-triazolo[4,3-*a*] pyridin-3(2*H*)-one, and 1*H*-benzotriazole were purchased from TCI Chemicals. 5-trifluoromethyl-2*H*-tetrazole was acquired from Flurochem. The remaining reagents were purchased from Sigma-Aldrich unless otherwise specified.

**Enzyme and protein assays**. β-Xylosidase activity was measured spectrophotometrically by the release of *p*-nitrophenol (*p*NP; $\varepsilon_{410} = 15{,}200\ \mathrm{M^{-1}\,cm^{-1}}$) from *p*-nitrophenyl β-D-xylopyranoside (*p*NPX). The standard reaction mixture contained 3.5 mM *p*NPX, 50 mM sodium acetate buffer (pH 5.0), and 0.1% bovine serum albumin (BSA) for stability purposes and the appropriate quantity of the purified enzyme in a final volume of 200 μL. Standard assays were incubated at 40 °C and 1200 rpm over 5 and 10 min to check the linearity of measured activity. Reactions were stopped by the addition of 500 μL 2% Na$_2$CO$_3$ (pH 11.7). The absorbance of controls without enzyme, prepared in the same conditions, was subtracted from the spectrometric measurements to avoid any overestimation of the activity due to autohydrolysis of *p*NPX. One unit of β-xylosidase activity is defined as the amount of enzyme necessary for the hydrolysis of 1 μmol of *p*NPX per min in the conditions specified for the assay. Reported measurements were taken from individual samples.

The bicinchoninic acid method was selected for protein quantification by using the Pierce™ Protein Assay Kit (Thermo Fisher Scientific) and following the manufacturer's instructions with BSA as the standard.

**Mutagenesis of BxTW1**. The acid/base catalytic residue of the β-xylosidase from *T. amestolkiae*, BxTW1[34], was identified through alignment with the GH3-enzyme JMB19063 (pdb: 3U4A)[80]. The β-xylosidase Xyl3A from *Hypocrea jecorina* (pdb: 5A7M) was selected by the SWISS-MODEL tool (https://swissmodel.expasy.org/) as the closest glycosidase to BxTW1 to generate the corresponding model.

Primers for site-directed mutagenesis were designed based on the nucleotide sequence of *bxtw1* (GenBank accession no. KP119719). Design was performed automatically using PrimerX (http://www.bioinformatics.org/primerx/) and the obtained sequences were as follows: E495A Fw 5′-GACAATACCATTGAAGCCG C̲TGGTACCGATCGTTTGAATG-3′ and E495A Rv 5′- CATTCAAACGATCGGT A̲CCAGCGGCTTCAATGGTATTGTC-3′; E495G Fw 5′- GACAATACCATTGA AGCC̲G̲GTGGTACCGATCGTTTGAATG-3′ and E495G Rv 5′ CATTCAAAC GATCG̲GTACCA̲CCGGCTTCAATGGTATTGTC-3′; E495Q Fw 5′-CAATAC CATTGAAGCC̲CAAGGTACCGATCGTTTG-3′; and E495Q Rv 5′- CAAACGA TCGGTACCT̲T̲GGGCTTCAATGGTATTG-3′. The pPICW1N (pPIC9:bxtw1) construct[32] was submitted to site-directed mutagenesis and amplified by PCR. The reaction mixtures contained 50 ng of pPICW1N plasmid DNA as template, 2U of Phusion High-Fidelity DNA Polymerase (Thermo Fisher Scientific), and 10 μL PCR buffer (5X Phusion HF Buffer from Thermo Fisher) in a final volume of 50 μL. Reaction mixtures were denatured at 98 °C for 30 s and then subjected to 18 cycles of amplification, each at 98 °C for 10 s, 58 °C for 20 s, and 72 °C for 5.5 min, followed by a final extension step at 72 °C for 10 min. The PCR product was digested by D*pn*I (New England Biolabs) in order to hydrolyze the parental DNA and then used to transform chemically competent *E. coli* DH5α cells. The sequence-verified plasmids were transformed into electrocompetent *P. pastoris* GS115 cells, after linearization with *Sac*I (Thermo Fisher Scientific). Transformed colonies were grown on Yeast Nitrogen Base plates in the absence of histidine as selection marker.

**Expression and purification of rBxTW1 wild-type and mutants**. Six clones from each mutant were screened for maximal protein production. The colonies were cultured in 50-mL tubes with 10 mL of YEPS medium (10 g/L yeast extract, 20 g/L peptone, 10 g/L sorbitol, and 100 mM potassium phosphate buffer pH 6.0). Cultures were incubated at 28 °C and 250 rpm and 5 g/L methanol were added daily as inducer of gene expression. A control with non-transformed *P. pastoris* GS115 was included. Enzyme production by rBxTW1 mutants was evaluated by loading the supernatants of 5-day-old cultures onto a 10% SDS-PAGE gel and comparing the intensity of the band corresponding to the expected molar mass of BxTW1. The total quantity of loaded protein was the same for all samples.

The *P. pastoris* GS115 strain expressing the wild-type rBxTW1[32] together with the selected clones producing the corresponding mutants were grown overnight in 250-mL flasks with 50 mL of YEPS medium at 28 °C and 250 rpm to prepare fresh inocula. Production of the enzymes was carried out using 3.5 mL of the inocula to start 100 mL cultures of YEPS medium in 1-L flasks. Conditions were set at 28 °C and 250 rpm with daily addition of 5 g/L methanol for 10 days. Cultures were harvested and centrifuged at 10,000 × *g* and 4 °C for 20 min. The corresponding supernatants were subsequently filtered using 0.22-μm disc filters (Merck-Millipore), ultrafiltered through a 50-kDa cutoff membrane (Merck-Millipore) and dialyzed against 10 mM acetate buffer (pH 4).

The wild-type rBxTW1 was purified by a single step of fast protein liquid chromatography using an ÄKTA Purifier system (GE Healthcare). The dialyzed crude was loaded onto a 5-mL HiTrap SPFF cartridge (GE Healthcare), equilibrated in 10 mM sodium acetate buffer (pH 4) and keeping a flow rate of 1 mL/min. Elution was carried out by applying a linear gradient of 1 M NaCl from 0 to 50% in 25 min. Fractions with β-xylosidase activity were collected and pooled together to be dialyzed and concentrated by ultrafiltration using 50-kDa cutoff Amicon Ultra-15 centrifugal devices (Merck-Millipore). The same procedure was carried out for the purification of the corresponding mutant versions of rBxTW1. The purified enzymes were stored at 4 °C.

**Screening of acceptors**. A library of potential acceptors was tested in glycosylation reactions catalyzed by the thioglycoligase variant rBxTW1-E495A. All samples contained 40 mM *p*NPX as donor and 4.0 g/L rBxTW1-E495A and assays were carried out at 40 °C and 1200 rpm for 1 h. Reactions were stopped by heating at 100 °C for 5 min, except for those acceptors whose potential products were considered highly labile: formic acid, L-glycine, L-glutamic acid, *N*-acetyl-L-glutamic acid, *N*-hydroxysuccinimide, and phenylphosphonic acid. In those cases, the reactions were stopped in ice and immediately analyzed by TLC. Acceptors were tested at a final concentration of 100 mM whenever possible. This value was decreased when enzyme inhibition was observed. Stocks of acceptors showing low solubility were prepared by adding an appropriate organic co-solvent. The content of co-solvent in the reaction was set at 10% (v/v) for ethanol (EtOH) and methanol (MetOH) and at 5% (v/v) for acetonitrile (ACN) and dimethyl sulfoxide (DMSO) to prevent the inhibition of rBxTW1-E495A. In the particular case of 5-(5-bromo-2-thienyl)-1*H*-tetrazole, the sample included *N*,*N*-dimethylformamide at a final

concentration of 7% (v/v), as the commercial reagent is provided as a solution in this solvent. Reactions were performed without buffer to avoid any possible competition with the assayed acceptor. However, when necessary, the initial pH was checked and roughly adjusted to a value close to the p$K_a$ of the acceptor by using pH-indicator strips pH 0–14 (Merck-Millipore), or with higher accuracy using a pH-meter if a sufficient quantity of the acceptor was available. The pH was adjusted in the range of 2.0–5.5, in which the enzyme remains active and stable, by adding HCl or NaOH. The formation of a xylosylated product was followed by TLC, selecting the eluent according to the expected polarity of the product. The conditions of this screening are summarized in Supplementary Data File 1.

**Ligase activity response assays**. Ligase activity response assays were carried out against a selection of potential acceptors encompassing a wide p$K_a$ range. The rationale of the assay is displayed in the Supplementary Fig. 10. Positive acceptors speed up the cleavage of the glycosyl-enzyme intermediate, which is the rate-limiting step of the whole reaction. The rupture of this intermediate occurs through a $S_N2$ reaction, forming a new glycosidic bond between the acceptor and the carbohydrate moiety, which is called ligase activity in this work. Regardless the stability of the newly synthesized glycoside in the conditions of the assay (and then of its detectability) the reaction will get faster because the free enzyme is regenerated more quickly. This results in rapid cleavage of the pNPX from donor, leading to increased release of pNP. This by-product is measured spectrophotometrically to determine the extent of the acceleration of enzyme regeneration.

In addition to the thioglycoligase, the wild-type enzyme was submitted to the same assays to serve as a control. Since the wild-type rBxTW1 possesses a functional acid/base catalyst, no acceleration of the reaction is expected, allowing the detection of any other possible effect of the acceptor on the enzymatic activity. Tested acceptors comprised ethanesulfonic acid, phenylphosphonic acid, benzenesulfinic acid, and formic acid at pH 2.2 using 50 mM glycine-HCl; benzeneseleninic acid, hydrazoic acid, and 2-(N-morpholino)ethanesulfonic acid at pH 3 using 50 mM glycine-HCl; N-hydroxysuccinimide, vanillin, 1H-1,2,3-triazolo [4,5-b]pyridine, 1H-1,2,3-triazole, acetophenone oxime, and D-glucose at pH 5.5 using 50 mM MES buffer. Reactions were performed as standard with constant donor concentration (3.5 mM pNPX) in the indicated buffer modifications. Increasing quantities of the acceptor were added up to a maximum concentration of 200 mM when possible, depending of acceptor solubility, possible enzyme inhibition and/or interference with the measurement of the released pNP. In any case, all compounds were tested up to a minimum of 20 mM, which was the concentration used in the glycosylation yield assays. Co-solvent was added when necessary as indicated in Supplementary Data File 1 and used at the same concentration in all conditions tested.

**Analysis of glycosylation yield**. Production yield of xylosides catalyzed by rBxTW1-E495A was studied using a selection of representative acceptors from each group of tested compounds: trans-cinnamic acid, vanillin, EGCG, N-hydroxybenzotriazole, 3,5-dibromo-1,2,4-triazole, thiophenol, benzenesulfinic acid, benzeneselenol, and phosphoric acid. Reaction conditions are displayed in Supplementary Table 4.

Co-solvent, when present, was at a concentration of 10% (v/v) and the production of each xyloside was monitored by HPLC over time up to a maximum of 4 h, comparing the effect of adding a suitable buffer versus simply adjusting the initial pH. The thioglycoligase was added at a concentration of 4.0 g/L and assays were carried out at 40 °C and 1200 rpm. Buffers were added at 50 mM except for N-hydroxybenzotriazole and 3,5-dibromo-1,2,4-triazole, for which it was increased to 100 mM because the desired pH value was near the limit of their buffering capacity. Yields were determined by HPLC as described below and expressed in terms of acceptor conversion (%). The former approach was slightly modified in the case of phosphoric acid. For this compound, the conversion rate of the donor pNPX was followed as the acceptor phosphoric acid was used in excess. In addition, the formation of the phosphoester was followed by NMR instead of HPLC. To ensure the monitoring of the initial stages of the reaction, enzyme concentration was decreased to 1 g/L, assays were carried out at 37 °C, and production was evaluated by recording spectra at different time intervals for 24 h.

**Analysis of the relation between p$K_a$, pH, and glycosylation**. Some additional ligase activity response assays and determination of glycosylation yields were carried out comparing the effect of maintaining the reaction pH at 2.2 (50 mM glycine-HCl), 3 (50 mM glycine-HCl), and 5.5 (50 mM MES). The ligase activity response assay was performed as described above using 5-trifluoromethyl-2H-tetrazole, benzenesulfinic acid, vanillin, and 1H-1,2,3-triazolo[4,5-b]pyridine. Glycosylation yields were analyzed in terms of acceptor conversion using 20 mM as initial concentration. The selected acceptors were benzenesulfinic acid, 5-(5-bromo-2-thienyl)-1H-tetrazole, vanillin, and 1H-1,2,3-triazolo[4,5-b]pyridine.

**Production of arabinofuranosyl and arabinopyranosyl derivatives**. The ability of rBxTW1-E495A to use different donors was studied by replacing pNPX with 4-nitrophenyl α-L-arabinofuranoside (pNPAf) and α-L-arabinopyranoside (pNPAp).

Assays were carried out at 40 °C and 1200 rpm using 4.0 g/L rBxTW1-E495A, 20 mM vanillin, 40 mM donor, and 10% ethanol as co-solvent. Reaction time was

increased in comparison to the former screening assays to assess the formation of product at 4 and 16 h. Product formation was followed by TLC and ESI-MS. Additionally, the arabinopyranosyl derivative was purified by HPLC and subsequently its structure was confirmed by NMR.

The glycosylation yield of the vanillin arabinopyranoside was obtained under the same conditions described for the equivalent xylopyranoside but extending the reaction time to 16 h.

**TLC analysis**. TLC was used to detect the formation of xylosides from the tested acceptors and to perform a preliminary purification prior to the ESI-MS analysis. In some cases, TLC was scaled to purify the required amount of the desired product for its characterization by NMR.

Analytical TLCs were carried out by using silica gel G/UV254 polyester sheets, (0.2 mm thickness and 40 × 80 mm plate size) provided by Macherey–Nagel. Eluent mixtures were chosen for each acceptor based on the expected polarity of the produced xyloside in comparison to the rest of compounds in the reaction mixture. Three different eluents were used, designated by A, B, and C, following an order of increasing hydrophobicity. The composition of each solution was as follows: (A) 1-butanol/acetic acid/H$_2$O 3:1:1 (v/v); (B) ethyl acetate/methanol/H$_2$O 10:2:1 (v/v); and (C) chloroform/methanol/formic acid 50:10:1 (v/v). Visualization of pNP, pNPX, and any aromatic acceptor and/or new xyloside was performed under 254 nm UV light. Additionally, sugars and glycoconjugates incorporating non-near UV-active compounds were detected by immersing the plates in a solution of methanol/sulfuric acid 95:5 (v/v) and revealing by heating at 110 °C for 10 min. The results were evaluated by comparing the pattern of spots between the reaction, its control without enzyme, and another control containing xylose, pNPX, and pNP.

With the purpose of avoiding the interference of the reaction components on the ESI-MS analysis, most of the reaction mixtures were previously submitted to a semipreparative purification step by TLC, which was carried out according to Robyt[81] with minor variations. Eluents and TLC plates were the same as used in the analytical assays. Aromatic xylosides were detected under 254 nm UV light whenever possible, while the position of the other glycoconjugates was established by comparison to a parallel TLC performed under the same conditions and revealed as described above. The silica containing the product was scrapped off, added to a 1 mL-solution of ethanol/water 85:15 (v/v) and sonicated for enhancing the solubilization of adsorbed compounds. Extraction suspensions were centrifuged at 20,000 × g for 5 min and supernatants were dried by speed vacuum and redissolved in 50 μL 10 mM sodium acetate buffer (pH 5). Reaction mixtures including silibinin, caffeic acid, and sinapic acid were submitted directly to ESI-MS analysis without prior semipreparative TLC with the purpose of detecting any possible secondary di- or tri-xylosylation.

Preparative TLC was selected as a rapid approach to purify products without an aromatic group, which cannot be monitored by absorbance in the near- or middle-UV range. The chromatography was carried out as explained above but using larger plates (silica gel 60 F$_{254}$ aluminum sheets, 0.2 mm thickness, and 20 × 20 cm plate size) provided by Merck. The product band was identified by comparison with a 5-mm wide strip cut out from the same plate and revealed as explained above. The scrapped silica containing the product band was suspended with 5-mL ethanol/water 85:15 (v/v) and sonicated. Extraction suspensions were centrifuged at 20,000 × g for 10 min and supernatants were dried by speed vacuum and redissolved in 500 μL of deuterated water. This protocol was applied for purifying the xylosides from hexanoic, heptanoic, phosphoric, and N-acetyl-L-glutamic acids. From the latter, two products were purified, the one linked through the δ-carboxyl group (compound 2) and the one hypothetically bond to the α-carboxyl group. Typically, the desired product was purified from 400 μL-reaction mixture performed in the conditions described for the acceptor screening (40 °C and 1200 rpm for 1 h, 40 mM pNPX and the appropriate concentration of acceptor and co-solvent). A total volume of 200 μL was loaded per sheet and two sheets were employed per reaction sample.

**HPLC analyses**. HPLC was used to monitor the synthesis of glycosides from thiophenol, trans-cinnamic acid, ferulic acid, p-coumaric acid, vanillin, EGCG, benzenesulfinic acid, benzeneselenol, N-hydroxybenzotriazole and 3,5-dibromo-1,2,4-triazole. Calibration curves were calculated for the acceptors and then used for quantifying both their consumption and the formation of the respective derived glycosides based on the areas under the corresponding peaks. A calibration curve for pNPX was also obtained to quantify the donor consumption when needed. HPLC was also used in a semipreparative fashion to purify the xylosides derived from the aforementioned acceptors together with hydrazoic acid and the arabinopyranoside from vanillin. Typically, the desired product was purified from 1 mL of a reaction mixture processed in the conditions described for the corresponding acceptor screening (40 °C and 1200 rpm for 1 h, 40 mM pNPX and the appropriate concentration of acceptor and co-solvent).

For compounds with UV absorbance, the HPLC analyses were performed on an Agilent 1200 series LC instrument, equipped with an UV-visible detector; in the case of xylosyl derivative of hydrazoic acid the analysis was performed on an Agilent 1100 series quaternary HPLC pump with an evaporative light-scattering detector ELSD-2000ES (Alltech). Depending of the product, different columns C18 or NH$_2$ were used: Kromasil 100-5-NH$_2$ column (4.6 × 250 mm) from Kromasil;

Mediterranea sea[18] TR-010006 semipreparative column (4.6 × 250 mm) from Teknokroma; and Eclipse Plus C18 column (4.6 × 250 mm) and ZORBAX Eclipse XDB-C18 column (4.6 × 150 mm) from Agilent. A summary of the protocol followed for each aromatic acceptor analyzed on the Agilent 1200 series LC instrument can be found in Supplementary Data File 2.

For purification of the xyloside from hydrazoic acid, the sample was loaded onto the Kromasil 100-5-NH$_2$ thermostated at 30 °C. The mobile phase was acetonitrile/H$_2$O 95:5 v/v (both containing 0.1% acetic acid) at 1 mL/min. Isocratic elution was carried out for 15 min. The product's peak was detected by light-scattering with the ELDS detector at 72.3 °C and 1.8 L/min of nitrogen gas. The fraction containing the product was collected by using a three-way flow splitter (model Accurate, from Dionex).

**NMR**. NMR experiments of the purified compounds were acquired at 25 °C (35 °C in the case of xylose 1-phosphate, compound 27) using a Bruker AVANCE 500 MHz spectrometer and a Bruker AVANCE 600 MHz spectrometer equipped with a cryogenic probe. Samples were prepared in 500 µL of deuterated water (D$_2$O). 1D $^1$H NMR spectra, $^1$H-$^{13}$C HSQC, HMBC, $^1$H-$^1$H TOCSY, and selective TOCSY experiments were carried out to fully assign the compounds. Corresponding pulse sequences included in TopSpin 2.1 Bruker software were employed.

For the kinetic and yield studies of xylose 1-phosphate different 1D $^1$H NMR spectra were recorded over time.

**Mass spectrometry analyses**. Samples were analyzed on a Bruker HCT Ultra ion trap and data were processed with DataAnalysis 3.4 (Bruker Daltonics), except for those containing caffeic and sinapic acids as acceptors. The latter were evaluated on a mass spectrometer with hybrid QTOF analyzer, model QSTAR, Pulsar I, from AB Sciex, and data were processed with Masshunter Data Acquisition B.05.01 and Masshunter Qualitative Analysis B.07.00 software (Agilent Technologies). Samples were introduced by direct infusion and ionized using electrospray with methanol as ionizing phase in positive and negative reflector mode.

**Reporting summary**. Further information on research design is available in the Nature Research Reporting Summary linked to this article.

## Data availability
Data supporting the findings of this work are available within the paper and its Supplementary Information files. Other data are available from the corresponding author upon reasonable request. Protein crystal structures were obtained from Protein Data Bank (PDB: 3U4A; PDB: 2X40; PDB :4I3G; PDB: 3ZYZ; PDB: 3AC0; and PDB: 5A7M). The nucleotide sequence of bxtw1 is publicly available at Genbank under the accession no. KP119719. Source data are provided with this paper.

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

## Acknowledgements

This work has been funded by Projects RTI2018-093683-B-I00, RTI2018-094751-B-C22, and PID2019-107476GB-I00 from MCIU/AEI/FEDER, UE, and S2018/EMT-4459 from Comunidad de Madrid. The authors thank Francisco J. Plou for his help with the purification of the xyloside from hydrazoic acid. The authors thank Nour Kayali for the ESI-MS analyses. B. Fdez. de Toro thanks to MCIU for a FPI fellowship and J.A. Méndez-Líter thanks the Tatiana Pérez de Guzmán el Bueno Foundation for his scholarship.

## Author contributions

M.N.D. and M.J.M. conceived the study. Z.A. and S.G.W. designed the obtention and the preliminary characterization of the mutant enzymes. M.N.D., A.G.S., F.J.C., S.G.W., and J.L.A. interpreted the data. M.N.D., L.I.E., L.B.M., and J.A.M.L. acquired the data except for NMR and ESI-MS spectra. B.F.T. acquired the NMR data. B.F.T. and F.J.C. identified the glycoconjugates from the NMR data. M.N.D. drafted the manuscript. A.P., L.I.E., A.G.S., J.A.M.L., and M.J.M. helped to draft, and substantially reviewed the manuscript.

## Competing interests

The authors declare no competing interests.

**Additional information**

