## [Peer Review File · Nature Communications]

REVIEWER COMMENTS

Reviewer #1 (Remarks to the Author):

The work reports the use of a highly versatile mutated enzyme for the synthesis of O-, S-, N-, Se-glycosides. In an overall manner, the manuscript is nicely written and the works largely deserve to be reported in Nature Communication. However I will still strongly advised to perform major corrections before acceptance.

Major Points:

1) please add missing references on thioligase or related topics:
Carbohydr Res. 2011 Sep 27;346(13):2028-33. doi: 10.1016/j.carres.2011.05.017
Beilstein J. Org. Chem. 2017, 13, 1857–1865. doi:10.3762/bjoc.13.180
Org. Biomol. Chem., 2011,9, 8371-8378 doi.org/10.1039/C1OB06227A
and others...

There are not so many references in the literature and they need all to be cited.

2) please also discuss the synthetic interest of this mutated thioligase towards C-, N- or S-glycosyltransferases. There are too many examples to be cited here, but this need to be discuss in lights of those enzymes. Please see also recent related work:
Molecular Catalysis Volume 479, December 2019, 110631,
<https://doi.org/10.1016/j.mcat.2019.110631>

3) only NRM Characterization were performed on O-glycosides. Please report NMR spectra for all compounds. Please also attribute each peaks, especially for C13. This is of outmost importance so to discriminate O-, S-, C-, N- or Se- glycosides.

4) please report at least one or 2 examples of synthetic interest with a scale up procedure. This is compulsory to assess the synthetic potency of such enzymatic procedure.

Reviewer #2 (Remarks to the Author):

In this work Nieto-Dominguez and colleagues report the use of a beta-xylosidase rBxTW1 mutant, BxTW1-E495A, from the ascomycete *Talaromyces amestolkiae*, as a potential thioglycoligase, in order to increase the enzymatic tool kit to synthesize a wide range of biological relevant glycoconjugates. rBxTW1 belongs to the GH3 family of glycoside hydrolases, showing the advantage to work at an optimal pH = 3, where the potential acceptors - e.g. carboxylic acids and phosphorylated derivatives - are neutral, avoiding potential electrostatic repulsion to enter into the catalytic site of the enzyme.

BxTW1-E495A shows an additional acid residue in its active site, besides the nucleophile and the acid/base residues, that accepts a proton from the acceptor, enhancing its nucleophilic capacity. The authors show that BxTW1-E495A is able to use a wide range of acceptor molecules for the production of O-/N-/S-/Se- xyalosides. Although others - Li et al. 2013/Pavic et al 2018 - described the use of phenols and carboxylic acids as acceptors using thioglycoligases, the authors perform an exhaustively study of the acceptors to be used by BxTW1-E495A. They carefully describe the influence of the pKa value of the acceptor in the reaction and determine a broad pKa value, from 2 to 9, for the reaction to take place. The authors concluded that rBxTW1-E495A can be considered as a proof of concept for a novel generation of multi-glycoligases obtained from the acidophilic members of the GH3 family. The manuscript is well written and makes a significant contribution to the field. Please, find below some comments/suggestions for improvement:

Major comments:

1. The abstract is missing.
2. The authors propose "enzymes from CAZy family GH3 as an ideal source of thioglycoligases. This family contains, amongst others, some glycosidases that are active at markedly low pH values. 3D structures of these enzyme indicated the presence of an additional acid residue in the active site (Supplementary, Fig. S1). Our hypothesis was that a thioglycoligase with these properties may be able to function in an acid environment instead of the conventional neutral/basic pH. Under these conditions, many potential acceptors, such as carboxylic acids and phosphorylated derivatives would be neutral and therefore able to enter the enzyme active site, thus avoiding any potential electrostatic repulsion which may limit the range of possible acceptors. Once there, the acceptor may transfer a proton to the secondary acid residue, becoming an enhanced nucleophile capable of attacking the glycosyl-enzyme intermediate. A comparison between the mechanism of wild-type retaining glycosidases, conventional thioglycoligases and that expected for such GH3 thioglycoligases is shown in Figure 1."

Is there any reported experimental data in the literature in which a mutant of the additional acidic residue shows reduction of the thioglycoside/hydrolase activity?

3. The authors study the activity of rBxTW1-E495A using a wide range of acceptor molecules, this data is mostly qualitative in Figure 2. What is the numeric meaning of the cross and the tick symbol on Table 1? Just detectable? Could the authors provide yields of the compounds identified by NMR in this table?
4. Could the authors compare the enzymatic activity of rBxTW1-E495A with other thioligases that are able to synthesize C-O-C linkages at neutral pH? Does rBxTW1-E495A have any kinetic advantages/disadvantages besides the use of a wide range of acceptor for working at acidic pHs, comparing with other enzymes that are able to work at neutral pHs?
5. The organization of the supplementary information is confusing. The first Table mentioned in the main text is Supplementary Table S6.

Minor comments:

Page 2, line 38: please remove ", etc".

Page 2, line 42: "catalytic site (12, 13)."

Page 3, line 63: please move the Supplementary Fig. 1 as Fig. 1A. The figure can be largely improved, e.g. by adding (i) residue numbers of structural homologues, (ii) a substrate/product visualized in the corresponding structural homologues.

Page 3, line 63: "of these enzymes"

Page 5, line 103: "rBxTW1-E495A"

Page 9, line 111: "their"

Page 15, line 197: please replace "E495A" by "rBxTW1-E495A" and along the entire manuscript.

Reviewer #3 (Remarks to the Author):

This a review about the manuscript NCOMMS-20-08993-T, titled "Active site mutation of fungal β -xylosidase from GH3 family reveals outstanding acceptor tolerance: From thio- to multi-glycoligases" by Nieto Domiguez et. al.

The manuscript deals with the production and characterisation of a glycoligase derived from a

fungal GH3 family xylosidase. The authors show that the ligase is exceptionally stable and active at a low pH regime, allowing substantially more different acceptors to be used in the reaction, which is shown with a wide variety of different acceptors carrying a range of functional reactive groups. The authors attempt to rationalise the reactivity using simple but effective metrics, like the pK_a of the reactive group.

The idea and concept proposed in the manuscript is interesting, but it remains to be seen if a simple characteristic of an enzyme, like the activity at low pH, is suitable enough to ensure a broad spectrum ligase as seen here. As an example the authors have taken a xylosidase, which is an inherent weakness of the paper, due to the rather limited number of biologically important glycosides. The collection of glycosides produced, seems also rather random, though it spans least the full range of possible pK_a 's.

It would have been much more interesting if some of the glycosides would be relevant for further use. This is indicated for some of them, but only in a side sentence. It might be interesting to expand on that with one or two sentences

In part the manuscript seems rushed, which shows in some parts as sloppy writing. Some of the sentences and paragraphs are nearly incomprehensible. Additional results and discussion, in the supplementary material would be really helpful for a general understanding of the concept and should be moved into the main section. That should be no problem in a pure online journal like Nature Communications.

Below, I will list in more detail some of my comments to different parts of the manuscript.

Title:

The title doesn't really reflect the content of the manuscript. The mutation itself has nothing to do with the outstanding acceptor tolerance of the enzyme.

Main:

Line 24 sequential  \diamond sequential

Line 24 to the assembly \diamond  for the assembly

Line 26 glicosciences  glycosciences

Line 28 These compounds  \diamond These conjugates

Line 29 molecules including  \diamond molecules, leading to#

Line 35 each class of  \diamond each type of

Line 38 one should mention carbohydrate-polymers of all sorts, That is what most GH's have as cognate substrate. Though understandably the focus is on glycoconjugates.

Line 43 ...the improvement of their synthetic.....

Line 45 to about 54: I think it should be logically the other way around. The retaining mechanism should be introduced first to understand the reason to mutate exactly this residue. I believe a reference to figure 1 would be timely here. Furthermore, the concept of thioglycoligase is introduced here for the first time and should be defined, taken into account that it derives from the special acceptor which usually has to be used.

Line 57 classified  \diamond classified

Line 57 120 families was quite a while ago. There are much more families now. One could update that number.

Line 61 GH3 family is proposed and than a xylosidase is chosen though there are enzymes acting on hexoses, which would have been more attractive. Of course there is the acceptor tolerance, but what is know of other enzymes in that class? Are they more specific in subsite +1

Furthermore one reason proposed is a the third acidic residue. Is there any plausible explanation pK_a that this residue could act as base to deprotonate the acceptor?

Line 98 Besides laboratory availability, which of the resulting glycosides might have functional relevance in a biological context?

Line 112: Thei r \diamond to many spaces typesetting?

Line 123 What is the thing with the activity of the enzyme? I don't understand in which context that plays a role here.

Line 128 What are alternative acceptors in that context?

Line 138 why may also? Is it for the both mentioned substances the solely likely explanation?
Line 139 to 141 That sentence is confusing. Rescue activity usually refers the break of the glycosylenzyme intermediate using a good acceptor. At this point the pnp is already cleaved off. How does one measure rescue activity via pnp release? Is a spontaneous decomposition taken into account and subtracted?

Line 150: does \diamond might?

Line 151 reference 15 seems not to state anything about the rapid degradation, just mentioned charged nucleophiles are bad for the activity. How does that fit in here?

Line 152 following: Thought I know what is meant with rescue activity, but wouldn't it be better to call it "ligase activity" what is here measured is the desired glycoside formation isn't it? Technically rescue assays for the wildtype aren't possible because the acid base is there. For which reason was that done? Just to check for possible inhibition competition with the natural hydrolysis or wildtype transglycosylation activity?

Line 153 testing \diamond tested

What is the purpose of the rescue assays performed in this context? It shows that the acceptors are reactive as shown before using TLC and ESI MS. There is a concentration dependence for some acceptors, but what else?

Line 178 Interestingly \diamond Interestingly

Line 180 conjugates \diamond conjugated

Line 190: to taking into account \diamond to take into account

To better understand the influence of the active site residues an estimate of the third residue in the active site would be very helpful to further rationalise the results.

The reactions to measure product yield are often at a pH close to the pKa as proposed earlier to be important. That would mean that despite remaining activity for substances with a rather high pKa would probably rarely lead to full conversion?

Line 246: derivatives \diamond derivatives

Line 261: Why was vanillin used for arabinofuranose. Vanillin wasn't the best acceptor for xylose either?

How good is arabinofuranose in the wildtype? How much would a lower K_m and therefore the binding and formation of the glycosylenzyme intermediate influence the reaction? Is the relative glycoside formation rate actually the same if the intermediate is actually formed?

Figures

Figure 3: some of the figures activities don't start at 0 activity, which is auto hydrolysis? Is the different starting activity solely due to the different reaction pH?

Figure 4: Similar issue. Here most certainly without acceptor the yield should be zero.?

Supplementary Material

Supplementary Results

It is mentioned that the third residue might involved in the deprotonation of the incoming receptor. Is that residue close enough to the +1 subsite or the bond to be formed/ cleaved to be of relevance. It looks too far away?

It would be interesting to discuss a bit the +1 subsite, which seems to show remarkable plasticity to accept so many different acceptors.

Line 440. That would mean acetic acid is an acceptor for the transglycosylation reaction. Isn't that just corroborating the results in the main part that you need MES or Glycine to have no influence of the buffer.

Line 445: same underlying mechanism. What is meant with that.

Line 456 to 460: I presume that was the first test, if the enzyme is able to act as thioglycoligase. It seems for the logical flow better to put that as a statement in the main section.

Supp Results 2.2 seems redundant, as this is discussed in the main section and is also supported by the effects with glutamate, described in the main section.

Supp Results 2.4 Again explains in more detail what is going on with other sugars and indeed propose the theory that the inherent lower affinity towards different sugars might be responsible for the different results.

That reinforces the perceived disconnection between the main and the supplementary part of the

manuscript. Some of the results in the main section could go into the supplementary and supplementary results should go into the main part to improve the logic flow and understandability of the manuscript.

REVIEWER COMMENTS

Response to the Reviewers' comments have been added in blue to facilitate the reading.

Reviewer #1 (Remarks to the Author):

The work reports the use of a highly versatile mutated enzyme for the synthesis of O-, S-, N-, Se-glycosides. In an overall manner, the manuscript is nicely written and the works largely deserve to be reported in Nature Communication. However I will still strongly advised to perform major corrections before acceptance.

We thank very much the reviewer for the positive assessment of our manuscript and for her/his valuable comments and suggestions. Detailed responses are given below.

Major Points:

1) please add missing references on thioligase or related topics:

Carbohydr Res. 2011 Sep 27;346(13):2028-33. doi: 10.1016/j.carres.2011.05.017

Beilstein J. Org. Chem. 2017, 13, 1857–1865. doi:10.3762/bjoc.13.180

Org. Biomol. Chem., 2011,9, 8371-8378 doi.org/10.1039/C1OB06227A

and others...

There are not so many references in the literature and they need all to be cited.

As the reviewer indicates, there is a limited number of reports on thioglycoligases. The first publication on every thioglycoligase mutant reported up to date, together with a reference showing the characterization of the corresponding wild-type enzyme, were compiled in Table S6 (Supplementary Information, page 24) and the main manuscript also contains reports on every thioglycoligase (page 4, line 90). Thanks for noticing the omission of the works by Almendros and coworkers (1) and Torpenholt and collaborators (2), they have been added in the revised version as well as Ati and coworkers (3), which has been incorporated into the introduction of the manuscript (line 54, page 3).

2) please also discuss the synthetic interest of this mutated thioligase towards C-, N- or S-glycosyltransferases. There are too many examples to be cited here, but this need to be discuss in lights of those enzymes. Please see also recent related work: Molecular Catalysis Volume 479, December 2019, 110631, <https://doi.org/10.1016/j.mcat.2019.110631>

We appreciate the reviewer's suggestion, since glycosyl transferases are the tool for glycoconjugate synthesis in nature. A brief commentary on the advantages and limitations of these enzymes in comparison to the thioglycoligase mutant, including the suggested reference, has been added (page 16, lines 383-392).

3) only NRM Characterization were performed on O-glycosides. Please report NMR spectra for all compounds. Please also attribute each peaks, especially for C13. This is of outmost importance so to discriminate O-, S-, C-, N- or Se- glycosides.

Both spectra and assignment tables of every identified glycoside have been incorporated to Supplementary Information (section 2.4).

4) please report at least one or 2 examples of synthetic interest with a scale up procedure. This is

compulsory to assess the synthetic potency of such enzymatic procedure.

In our opinion, among the diverse list of acceptors tested in the present work, the capacity of the mutant to selectively attach a xylose moiety to the most acidic hydroxyl of a flavonoid is currently the most interesting trait from a synthetic perspective. In this regard, it should be noticed that xylose, despite of its secondary role in mammalian physiology, is of the utmost importance in the plant kingdom, being a substituent of numberless natural plant metabolites. However, the role of these glycoconjugates is poorly understood due to the difficulties associated to their chemical synthesis, purification or even accurate identification (4-6). This is particularly important in the case of flavonoids, which are complex polyphenols present in low quantity and in different glycosylated forms in plant tissues. In addition, flavonoids are secondary metabolites not only involved in plant defense (7), but also associated to a plethora of benefits for human health (8).

In the present work three flavonoids with reported therapeutic activities have been successfully glycosylated: quercetin (9), silibinin (10, 11) and epigallocatechin gallate (12). The production of the latter xylosyl derivative was studied in more detail reaching a moderate yield of about 30% (Fig. 5) with complete selectivity (Fig. 2, compound **13**), which is a quite remarkable result since this acceptor is a bulky polyphenol with eight potential glycosylable positions. How glycosylation affects the properties of flavonoids is hardly predictable despite the fact that these polyphenols are commonly found associated to carbohydrates (4). Nevertheless, our group and others have reported many cases in which the addition of a sugar moiety improves certain bioactivities, biosafety, stability and/or solubility of phenolic aglycons (13-15). More specifically, there are studies on the promising properties of glycosides from EGCG (16), silibinin (17) and quercetin (18). The last reference is in fact an analysis of the therapeutic potential of a quercetin xylosyl derivative.

Beyond the three flavonoids tested in the present work, there are many other potential members of this class of polyphenols with known biological activities fulfilling the pK_a requirements of the enzyme, which are therefore expected to be glycosylated following the procedure established for the former flavonoids. Among them, catechin and vitexin are particularly interesting since they are known to possess bioactive xylosyl derivatives (19, 20). On the other hand, natural xylosides of kaempferol (21), cyanidin (22) or luteolin (23) have been identified, although their function or potential pharmacological activities remain unknown.

Regarding the scale-up procedure, the recombinant enzyme is profusely secreted by *Pichia pastoris* and purified in one step, producing considerable amounts of biocatalyst (24). However, the reduced specific activity of this class of mutants increases the enzyme cost. To develop a more cost-effective process we are currently working in the immobilization of rBxTW1-E495A, and preliminary results indicate that once attached to an insoluble carrier the mutant keeps its synthetic capabilities.

Once the procedure for immobilizing rBxTW1-E495A is well established, the next step will be to scale-up the synthesis of one of the aforementioned polyphenyl xylosides to produce an amount high enough for analyzing its biological properties.

A brief commentary on the synthetic importance of glycosylating flavonoids and the necessity of reducing the biocatalyst cost has been added to the manuscript (page 12, lines 278-289; page 16, lines 392-394).

Reviewer #2 (Remarks to the Author):

In this work Nieto-Dominguez and colleagues report the use of a beta-xylosidase rBxTW1 mutant, BxTW1-E495A, from the ascomycete *Talaromyces amestolkiae*, as a potential thioglycoligase, in order to increase the enzymatic tool kit to synthesize a wide range of biological relevant glycoconjugates. rBxTW1 belongs to the GH3 family of glycoside hydrolases, showing the advantage to work at an optimal pH = 3, where the potential acceptors - e.g. carboxylic acids and phosphorylated derivatives - are neutral, avoiding potential electrostatic repulsion to enter into the catalytic site of the enzyme.

BxTW1-E495A shows an additional acid residue in its active site, besides the nucleophile and the acid/base residues, that accepts a proton from the acceptor, enhancing its nucleophilic capacity. The authors show that BxTW1-E495A is able to use a wide range of acceptor molecules for the production of O-/N-/S-/Se- xyalosides. Although others - Li et al. 2013/Pavic et al 2018 - described the use of phenols and carboxylic acids as acceptors using thioglycoligases, the authors perform an exhaustively study of the acceptors to be used by BxTW1-E495A. They carefully describe the influence of the pKa value of the acceptor in the reaction and determine a broad pKa value, from 2 to 9, for the reaction to take place. The authors concluded that rBxTW1-E495A can be considered as a proof of concept for a novel generation of multi-glycoligases obtained from the acidophilic members of the GH3 family. The manuscript is well written and makes a significant contribution to the field. Please, find below some comments/suggestions for improvement:

We really appreciate the reviewer's positive opinion of our work, and acknowledge the suggested corrections that will certainly improve the quality of the manuscript.

Major comments:

1. The abstract is missing.

We sincerely apologize for the omission. The abstract section has been added to the main manuscript (pages 1-2, lines 19-32). We will carefully review that all the necessary files are uploaded when submitting the corrected version.

2. The authors propose "enzymes from CAZy family GH3 as an ideal source of thioglycoligases. This family contains, amongst others, some glycosidases that are active at markedly low pH values. 3D structures of these enzyme indicated the presence of an additional acid residue in the active site (Supplementary, Fig. S1). Our hypothesis was that a thioglycoligase with these properties may be able to function in an acid environment instead of the conventional neutral/basic pH. Under these conditions, many potential acceptors, such as carboxylic acids and phosphorylated derivatives would be neutral and therefore able to enter the enzyme active site, thus avoiding any potential electrostatic repulsion which may limit the range of possible acceptors. Once there, the acceptor may transfer a proton to the secondary acid residue, becoming an enhanced nucleophile capable of attacking the glycosyl-enzyme intermediate. A comparison between the mechanism of wild-type retaining glycosidases, conventional

thioglycoligases and that expected for such GH3 thioglycoligases is shown in Figure 1.”

Is there any reported experimental data in the literature in which a mutant of the additional acidic residue shows reduction of the thioglycoside/hydrolase activity?

Unlike other families, there are not many structure-function studies in GH3 enzymes. However, Pozzo and coworkers (25) described the presence of the additional acidic residue (D58) in the β -glucosidase 3B from *Thermotoga neapolitana*, and reported that it was conserved in at least 200 members of the family, posing its potential role in the accommodation of the substrate at the -1 subsite. More interestingly, mutation of this residue caused a dramatic decrease in k_{cat} , although the hydrolytic activity was partially restored by adding azide. These results, together with the significant residual activity observed when the catalytic acid/base was mutated, led the authors to suggest the implication of the additional acidic residue in the catalytic mechanism.

Li and coworkers (26) also mutated the additional acid residue (D71) during the identification of the acid/base catalyst of the β -glucosidase fbgl from *Flavobacterium meningosepticum*. In this case, no hypothesis was formulated for the role of this residue, but kinetic analyses displayed a remarkable loss of k_{cat} for the mutants at D71, reaching values as low as those determined for the mutants at the acid/base catalyst position.

The conserved additional acid residue was identified in other studies, for instance as D98 in β -glucosidase DesR from *Streptomyces venezuelae* (27), as D95 in β -D-glucan exohydrolase ExoI from *Hordeum vulgare* (28), as E125 in soprimeverose-producing enzyme from *Aspergillus oryzae* (29), as E107 in β -xylosidase XlnD from *Aspergillus oryzae* (30) or as D109 in exo-1,3/1,4-beta-glucanase EXOP from *Pseudoalteromonas* sp. BB1 (31). However, no mutants were characterized and only its role in substrate accommodation was stated.

A brief comment on the available information about this residue has been added to the manuscript (pages 4-5, lines 97-101).

3. The authors study the activity of rBxTW1-E495A using a wide range of acceptor molecules, this data is mostly qualitative in Figure 2. What is the numeric meaning of the cross and the tick symbol on Table 1? Just detectable? Could the authors provide yields of the compounds identified by NMR in this table?

The products detected by the technique indicated in the header of the column are marked with a tick, and those not detected with a cross. This information has been added to the corresponding footnote (page 24, line 554). We apologize for the omission. In addition, the maximal conversion yield values, when determined, have been incorporated into Table 1 (page 23-24) as requested.

4. Could the authors compare the enzymatic activity of rBxTW1-E495A with other thioglycoligases that are able to synthesize C-O-C linkages at neutral pH? Does rBxTW1-E495A have any kinetic advantages/disadvantages besides the use of a wide range of acceptor for working at acidic pHs, comparing with other enzymes that are able to work at neutral pHs?

The comparison suggested by the reviewer is very interesting, although it is difficult to elaborate a general response since, to the best of our knowledge, only two other thioglycoligases able to form C-O-C linkages have been reported up to date.

These enzymes are the α -xylosidase YicI from *E. coli* (32, 33) and the cyclodextrin glucanotransferase Cgt from *Bacillus* sp. I-5 (34, 35) and their ability to glycosylate hydroxyl acceptors was explained by the following rationale: both biocatalysts are retaining α -glycosidases and therefore the glycosyl-enzyme intermediate is established with a β -configuration, which is intrinsically more reactive than the α -glycosyl-enzyme intermediates typical of retaining β -glycosidases, and so this extra instability may explain why these enzymes do not require strong nucleophiles (they generally use thiols) to disrupt the intermediate, being able to use alcohols instead.

When comparing the performance of these enzymes and rBxTW1-E495A, it is worth noting that YicI and Cgt employ fluorinated sugar substrates as donors, whereas rBxTW1-E495A uses a *p*-nitrophenyl derivative (*p*NPX). In the case of YicI, it is specifically stated that the enzyme is two orders of magnitude faster using α XylF in comparison to *p*NP α Xyl. The preference is not specified for Cgt, but as α G2F is used as donor and nitrophenyl maltosides as acceptors, it can be assumed the enzyme is highly inefficient hydrolyzing the latter. The requirement of fluorinated sugar donors can be considered disadvantageous since most of them are not commercially available and, more importantly, they are more sensible to longer reaction times and alkaline pH and yields can be deleteriously affected simply by spontaneous hydrolysis of the donor (34). In this sense, both working at acidic pH and using a *p*-nitrophenyl sugar substrate should increase stability for donor and potential products, providing more freedom to adjust reaction time and enzyme dosage. This is especially important in the case of xylosyl esters (also C-O-C products) which are more sensitive to neutral pH than conventional glycosides, as it has been experimentally corroborated in this work with the xylosyl derivative of *trans*-cinnamic acid (Supplementary Information, page 30, Fig. S5).

From the point of view of the reaction yield, rBxTW1-E495A glycosylated 66% of vanillin, 79% of *p*-coumaric acid at the hydroxyl position, 82.7% of ferulic acid at the hydroxyl position, and 31% of EGCG. Cgt requires high donor excess, otherwise modest conversion values (22-36%) are attained (34, 35). The activity of the YicI mutant seems to be limited to certain nitrophenols and *p*-nitrophenyl sugars, reaching high conversion yields with the suitable compounds and low or even no conversion for the rest of them, which indicates a rather restricted acceptor range (32, 33).

However, it should be stated that, apart from the stabilizing effect of moderately low pH values, the reported differences are probably specifically related to the selected enzyme rather than the basic catalytic mechanism. It would not be surprising if other thioglycoligases from the families of rBxTW1, YicI and Cgt behaved differently in terms of yield, acceptor specificity or preferred donors.

A brief commentary highlighting the practical advantages of operating in the acidic pH range has been added to the manuscript (page 15, lines 351-358).

5. The organization of the supplementary information is confusing. The first Table mentioned in the main text is Supplementary Table S6.

Prior to the section of additional results, Supplementary contains detailed information on material and methods that is supported by several figures and tables. However, the main text

refers Supplementary mainly to mention supporting results, this being the reason why Table S6 is the first one cited. Nevertheless, attending to reviewer's suggestion and also comments from Reviewer #3, additional information has been incorporated into the manuscript, previous to the main results. Supplementary information on the preliminary characterization of the mutant and full data from NMR and ESI-MS was briefly referred to and summarized in order to improve clarity (page 6, lines 133-146; page 7, lines 155-157).

Minor comments:

Page 2, line 38: please remove “, etc”. Corrected (page 3, line 54).

Page 2, line 42: “catalytic site (12, 13).” The sentence has been rewritten (page 3, lines 57-60).

Page 3, line 63: please move the Supplementary Fig. 1 as Fig. 1A. The figure can be largely improved, e.g. by adding (i) residue numbers of structural homologues, (ii) a substrate/product visualized in the corresponding structural homologues.

Changes have been addressed as suggested, and Fig S1 has been moved to the main text as Figs. 1B and 1C. Polar contacts between a β -D-glucose ligand and β -glucosidase from *Hypocrea jecorina* are indicated as reported by Karkehabadi and coworkers (36).

Page 3, line 63: “of these enzymes” The sentence has been rewritten (page 4, line 97-98).

Page 5, line 103: “rBxTW1-E495A” Corrected (page 23, line 553).

Page 9, line 111: “their” Corrected (page 25, line 582).

Page 15, line 197: please replace “E495A” by “rBxTW1-E495A” and along the entire manuscript. Corrected (page 11, line 262; page 11, line 268)

Reviewer #3 (Remarks to the Author):

This a review about the manuscript NCOMMS-20-08993-T, titled “Active site mutation of fungal β -xylosidase from GH3 family reveals outstanding acceptor tolerance: From thio- to multi-glycoligases” by Nieto Domiguez et. al.

The manuscript deals with the production and characterisation of a glycoligase derived from a fungal GH3 family xylosidase. The authors show that the ligase is exceptionally stable and active at a low pH regime, allowing substantially more different acceptors to be used in the reaction, which is shown with a wide variety of different acceptors carrying a range of functional reactive groups. The authors attempt to rationalise the reactivity using simple but effective metrics, like the pK_a of the reactive group.

The idea and concept proposed in the manuscript is interesting, but it remains to be seen if a simple characteristic of an enzyme, like the activity at low pH, is suitable enough to ensure a broad spectrum ligase as seen here. As an example the authors have taken a xylosidase, which is an inherent weakness of the paper, due to the rather limited number of biologically important glycosides. The collection of glycosides produced, seems also rather random, though it spans least the full range of possible pK_a 's.

It would have been much more interesting If some of the glycosides would be relevant for further use. This is indicated for some of them, but only in a side sentence. It might be interesting to expand on that with one or two sentences

In part the manuscript seems rushed, which shows in some parts as sloppy writing. Some of the sentences and paragraphs are nearly incomprehensible. Additional results and discussion, in the supplementary material would be really helpful for a general understanding of the concept and should be moved into the main section. That should be no problem in a pure online journal like Nature Communications.

Below, I will list in more detail some of my comments to different parts of the manuscript. We are thankful for the thorough review of our manuscript. We have carefully gone over the text following Reviewer's comments, questions, and suggestions. We have paid especial attention to improve writing and to clarify the reasoning and purpose of the assays performed. Detailed answers to the comments are displayed below.

Title:

The title doesn't really reflect the content of the manuscript. The mutation itself has nothing to do with the outstanding acceptor tolerance of the enzyme.

We concur with the reviewer's comment in that the mutation is not probably the ultimate reason for the enzyme promiscuity, although it was necessary to generate the thioglycoligase mutant, which revealed the vast acceptor range. We have modified the title to *Thioglycoligase derived from fungal GH3 β -xylosidase reveals outstanding acceptor tolerance: From thio- to multi-glycoligases*.

Main:

Line 24 secuential  \diamond sequential. Corrected (page 2, line 39).

Line 24 to the assembly \diamond  for the assembly. Corrected (page 2, line 40).

Line 26 glicosciences  glycosciences. Corrected (page 2, line 42).

Line 28 Theses compounds  \diamond These conjugates. Corrected (page 2, line 44).

Line 29 molecules including  \diamond molecules, leading to# Corrected (page 2, line 45).

Line 35 each class of  \diamond each type of Corrected (page 3, line 51).

Line 38 one should mention carbohydrate-polymers of all sorts, That is what most GH's have as cognate substrate. Though understandably the focus is on glycoconjugates.

A mention of carbohydrate-polymers, indicating their role as the main substrates of GHs, has been added to the manuscript (page 3, lines 57-60).

Line 43 ...the improvement of their synthetic..... The sentence has been rewritten (page 3, lines 57-60).

Line 45 to about 54: I think it should be logically the other way around. The retaining mechanism should be introduced first to understand the reason to mutate exactly this residue. I believe a reference to figure 1 would be timely here. Furthermore, the concept of thioglycoligase is introduce here for the first time and should be defined, taken into account that it derives from the special acceptor which usually has to be used.

We have modified the explanation to firstly introduce the conventional mechanism of retaining glycosidases since we understand that the broad audience of Nature Communications may not be familiar with this particular enzyme class (pages 3-4, lines 61-82). The modified text includes the specific mention of thiols as acceptors and thioglycosides as products.

Line 57 clasified  \diamond classified. Corrected (page 4, line 88).

Line 57 120 families was quite a while ago. There are much more families now. One could

update that number. Corrected (page 4, line 88)

Line 61 GH3 family is proposed and than a xylosidase is chosen though there are enzymes acting on hexoses, which would have been more attractive. Of course there is the acceptor tolerance, but what is know of other enzymes in that class? Are they more specific in subsite +1 Furthermore one reason proposed is a the third acidic residue. Is there any plausible explanation pKa that this residue could act as base to deprotonate the acceptor?

The GH3 family contains many examples of β -glucosidases, which are indeed interesting candidates to be exploited as thioglycoligases. However, we also considered β -xylosidases as candidates of utmost importance due to the central role of xylose in the plant kingdom. The potential interest of the synthesized xylosides or other related xylosyl derivatives is discussed in response to the next comment and to question 4 from Reviewer #1. In addition to its synthetic potential and the previously known versatility of the wild-type rBxTW1 catalyzing transxylosylation reactions, there were also some technical advantages in using this β -xylosidase. rBxTW1 has a remarkably low optimum pH, which allowed using uncommon acceptor groups at their neutral state, such as sulfinic, phosphonic and phosphoric acids (page 5, lines 101-106). Moreover, due to the absence of ramifications in pentopyranoses, *p*NPX is intrinsically more reactive than its hexose derived counterparts. By doing so, the use of dinitrophenyl sugar donors was avoided, an unquestionable advantage considering that they are much more unstable and difficult to handle (37). A brief comment summarizing these advantages has been added to the main text (page 6, lines 128-131).

Regarding acceptor tolerance, unfortunately it is not a property routinely assayed in conventional characterization of glycosidases, and when analyzed, the focus is usually on the synthesis of oligosaccharides (38, 39) or alkyl glycosides (40). As far as we know, no other enzyme from the GH3 family has been reported to display an acceptor tolerance similar to rBxTW1 (24). From the structural perspective, Suzuki and coworkers (41) compared a series of solved structures of GH3 glycosidases and defined a triad of aromatic residues (W-F-Y) as important for substrate positioning in the +1 subsite; in a more recent report (42), this triad was found to be a general feature of fungal GH3 β -glucosidases. However, the rationales about subsite +1 are largely restricted to explain the enzyme specificities for the different natural substrates (both carbohydrates and glycoconjugates) (29, 36, 42-47). Seidle and coworkers (48) associated one of these residues to the hydrolysis/transglycosylation ratio for a GH3 β -glucosidase from *Aspergillus niger*, but we have not found reports discussing structural insights of subsite +1 in regard to tolerance of glycosylation acceptors.

The reviewer also asks about the deprotonation of the acceptor by the third acid residue. In a conventional transglycosylation reaction catalyzed by wild-type glycosidases, the catalytic acid/base residue receives a proton from the acceptor. Conversely, thioglycoligases require strong nucleophiles as acceptors since they are able to attack without catalytic assistance. According to our hypothesis, the third acid residue could mimic the function of the mutated catalytic acid/base, but such mutation greatly loosens the constraints for acceptor accommodation allowing the diversity reported in the manuscript. As discussed below, obtained results suggest that the acceptor must enter in the neutral form and then it is deprotonated in the active site. In this scenario, E91 seems to be the most suitable candidate to assist such deprotonation.

Line 98 Besides laboratory availability, which of the resulting glycosides might have functional relevance in a biological context?

The list of selected acceptors was elaborated mainly under the criteria of having a wide diversity of potentially glycosylable functional groups and covering as much as possible the values of the defined pK_a range.

However, biological relevance was also considered. In this regard, and as explained above (Reviewer #1, comment 4), xylose is a particularly important sugar in the plant kingdom and xylosyl derivatives of flavonoids are the most promising glycosides synthesized by rBxTW1-E495A to be relevant for both plant physiology research and pharmaceutical applications (page 12, lines 278-289). From a general perspective, the effect of attaching a xylose moiety (or any other carbohydrate) is hardly predictable, but we can summarize some of the potential outcomes based on similar glycosides described in other works.

Enzymatically synthesized sugar fatty acid esters are considered a green alternative to petroleum-derived surfactants. Among other properties, they have been reported to be biodegradable, non-toxic, and antimicrobial compounds (49). Several examples of these glycoconjugates have been described, including some derived from short or medium chain length carboxylic acids (49-51) similar to the ones successfully tested as acceptors for rBxTW1-E495A (pages 23-24, Table 1). In fact Abdulmalek and coworkers (52) reported the synthesis of xylosyl hexanoate ester catalyzed by a lipase, and an equivalent ester was obtained in the present work using rBxTW1-E495 (compound **6**).

The data referring to glycosyl derivatives of triazoles and tetrazoles are more difficult to interpret due to the huge diversity of these aglycons. However, several studies have been published synthesizing a number of these derivatives, frequently assaying them for different bioactivities of interest (53-57). The work of Wilkinson and coworkers (57) is particularly interesting in this context, since an antimicrobial effect is reported for triazole derivatives of arabinose, a pentose similar to xylose.

Thioglycosides, selenoglycosides and glycosyl sulfones act as glycosidase inhibitors, a property conferring them different bioactivities of pharmacological interest (58-61). This includes also glycosides derived from aromatic acceptors (62, 63), as it is the case for the ones synthesized in the present work (compounds **29** and **31**). It should be noticed that in the work of Ayers and collaborators (59) the carbohydrate component of the sulfone derivative is again arabinose.

The potential applications of *O*-xylosyl phosphates or *O*-xylosyl phosphonates are less clear. Despite the fact that sugar 1-phosphates are products with great biotechnological potential, most of the studied compounds are α -sugars and when it comes to the β -anomers the available information on their biological roles is mainly restricted to β -glucose 1-phosphate (64). However, two GH3 wild-type retaining β -glycosidases have been recently reported to act as true glycoside phosphorylases. Both enzymes successfully catalyzed the synthesis of not only β -glucose 1-phosphate, but also β -*N*-acetylglucosamine 1-phosphate, suggesting that other 1-phosphorylated β -sugars might also have a biological function yet to be determined (65, 66).

A summarized version of the former response has been added to the main text (pages 12-13, lines 290-319).

Line 112: Thei r \diamond to many spaces typesetting? Corrected (page 25, line 582).

Line 123 What is the thing with the activity of the enzyme? I don't understand in which context that plays a role here.

Following the proposed rationale, acceptors with $pK_a \geq 10$ would require at least a reaction pH value about 7-8 to allow their partial deprotonation. rBxTW1 is not stable at those pH values and residual activity is very low even at short reaction times (24). Consequently, if the reaction is buffered at pH 7-8, rBxTW1-E495 would be highly impaired to catalyze the first step of the double displacement mechanism, necessary to reach the glycosyl-enzyme intermediate. Without this intermediate, the glycosylated product cannot be formed, regardless of whether the acceptor is deprotonated or not. On the contrary, if the reaction is buffered at a more acidic pH, rBxTW1-E495 would catalyze the formation of the glycosyl enzyme intermediate, but the acidic environment will prevent the required deprotonation of the acceptor. Thus, the reaction pH should be compatible not only with the deprotonation of the acceptor but also with the activity of the enzyme. A brief explanation has been added to the manuscript in order to clarify this point (page 8, lines 177-183).

Line 128 What are alternative acceptors in that context?

We apologize for using the word "alternative" which may be confusing. It has been deleted in the corrected version of the manuscript (page 8 line 187). We simply meant those acceptors from Table 1 with pK_a values in the ranges 1-2 or 9-10.

Line 138 why may also? Is it for the both mentioned substances the solely likely explanation?

The interference of a formal charge in the proximity of the acceptor group is, in our opinion, the most likely explanation for the absence of detectable xyloside. However, the alternative of product instability was also considered, though discarded after carrying out the ligase activity response assays using L-glycine and MES. The sentence has been rewritten to clarify this reasoning (page 8 line, 194-197).

Line 139 to 141 That sentence is confusing. Rescue activity usually refers the break of the glycosyl-enzyme intermediate using a good acceptor. At this point the pnp is already cleaved off. How does one measure rescue activity via pnp release? Is a spontaneous decomposition taken into account and subtracted?

We agree on the fact that rescue activity refers to the disruption of the glycosyl-enzyme intermediate. However, this step is necessary to regenerate the ability of the enzyme to catalyze the cleavage of a new molecule of *p*NPX. In the case of thioglycoligases, the formation of the glycosyl-enzyme intermediate is faster than its disruption, when a donor with a relatively good leaving group (e.g. *p*NP) is used (67). This fact, together with the high concentration of *p*NPX in comparison to rBxTW1-E495A implies that in practical terms the deglycosylation of the enzyme is the rate-limiting step for the release of *p*NP.

Spontaneous decomposition was measured using controls without biocatalyst. The absorbance of these controls is subtracted to the final activity (Supplementary, page 3, lines 75-77), but it is usually a negligible value. However, the enzyme keeps the ability of cleaving the glycosyl

intermediate at a low rate when no nucleophile is added. This is considered by showing the activity value when concentration of acceptor is zero.

Reference 15 (renumbered as 16 in the corrected manuscript) (68), whose suitability is discussed below, shows a similar assay studying the influence on enzymatic activity of adding different nucleophiles to a glycosidase mutant in the acid/base catalyst. This effect is measured through the release of 2,4-dinitrophenol. Kinetic studies reported in the cited work show that, in the absence of nucleophiles, the enzyme is still able to catalyze the deglycosylation step, although k_{cat} values with the addition of suitable acceptors are several orders of magnitude higher. A brief explanation has been added to the manuscript in order to clarify this topic (page 9, lines 214-219) and also to the Supplementary Information (page 11, lines 182-192) together with an explanatory scheme (page 11 figure S1).

Line 150: does \diamond might?

We apologize for the confusing writing. The reasoning is that the product might form but being too unstable to accumulate it would not be detected. The sentence has been modified to present this idea more clearly (page 9, line 208-211).

Line 151 reference 15 seems not to state anything about the rapid degradation, just mentioned charged nucleophiles are bad for the activity. How does that fit in here?

Reference 15 (renumbered as 16 in the corrected manuscript) (68) has been included because, in our opinion, the reported methodology and conclusions support the logic of the present work.

Wang and coworkers carried out the identification of the acid/base catalyst of the retaining β -glucosidase Abg. The activity of the E170G mutant was studied in the presence of increasing concentrations of different exogenous nucleophiles using 2,4-dinitrophenyl- β -D-glucopyranoside (2,4-DNPG) as substrate, an approach similar to the ligase activity response assays performed in the present study. In the same way, the activity was measured in terms of released 2,4-DNP and some of the nucleophiles led to remarkable increases of k_{cat} . However, only the formation of β -glucosyl azide and the thiophenol adduct were confirmed, whereas the absence of products for those reactions containing acetate, benzoate or formate was explained as “possibly because of decomposition of labile acylal adducts”. It should be mentioned that in our study the formation of a xylosyl derivative from acetate was confirmed, probably because in the acidic conditions selected for reactions with rBxTW1-E495A this ester is more stable than in those reported in reference 15 (pH 7.0).

Line 152 following: Thought I know what is meant with rescue activity, but wouldn't it be better to call it “ligase activity” what is here measured is the desired glycoside formation isn't it?

Technically rescue assays for the wildtype aren't possible because the acid base is there. For which reason was that done? Just to check for possible inhibition competition with the natural hydrolysis or wildtype transglycosylation activity?

We agree with the observation that the assay on the wild-type enzyme is not properly a rescue activity experiment, although it is measured as the increase of enzyme activity. The name of the assay has been replaced throughout this work by “ligase activity response assay” based on this reviewer's suggestion. Regarding the role of the wild-type rBxTW1 in these experiments, it was included in the assay as a control to detect any potential effect of the acceptor different to the

nucleophilic attack on the glycosyl-enzyme intermediate, such as competitive inhibition or allosteric behavior (69-71). The manuscript (page 9, lines 219-222) and the Supplementary Information (pages 11-12, lines 201-204) have been modified to clarify the role of the wild-type enzyme in these experiments.

Line 153 testing \diamond tested. The sentence has been rewritten (page 9, lines 211-214).

What is the purpose of the rescue assays performed in this context? It shows that the acceptors are reactive as shown before using TLC and ESI MS. There is a concentration dependence for some acceptors, but what else?

Certainly, TLC and ESI-MS allow the detection of many reactive acceptors. However, none of these approaches distinguishes between unreactive acceptors and those for which the glycosylated product is unstable. To gain insight into the acceptor promiscuity of rBxTW1-E495A such differentiation is essential. The ligase activity response assays do make the distinction, since all the reactive acceptors will speed up the reaction rate regardless of the stability of the glycosylated product.

In the present work, these analyses led to determine that formic acid, benzeneseleninic acid and *N*-hydroxysuccinimide accelerated the cleavage of the glycosyl-enzyme intermediate in the same way as the rest of tested compounds within the pK_a range from 2 to 9 identified as suitable acceptors of rBxTW1-E495A. This occurred even though no xyloside was detected, neither by TLC nor by ESI-MS, for the three mentioned chemicals. Consequently, the ligase activity response assay supported the hypothesis of this study, indicating that the three compounds still behaved as expected and that the absence of the corresponding xylosyl derivatives was probably due to their instability. A brief sentence has been added to the manuscript to make this conclusion clearer (page 10, lines 233-236).

Line 178 Interestengly \diamond Interestingly. Corrected (page 10, line 242).

Line 180 conjugates \diamond conjugated. Corrected (page 10, line 244).

Line 190: to taking into account \diamond to take into account. The sentence has been rewritten (page 11, lines 253-254).

To better understand the influence of the active site residues an estimate of the third residue in the active site would be very helpful to further rationalise the results.

In response to a similar request from Reviewer #2, Figure 1 has been modified to include the superimposition of several enzyme crystal structures from family GH3 (Fig. 1B) and a model of rBxTW1 (Fig. 1C). The figure shows the catalytic nucleophile, the catalytic acid/base and the third conserved acid residue from the selected structures. Polar contacts with the three residues are displayed for a carbohydrate ligand from one of the solved structures. Figure 1C depicts the conserved residues from the active site of rBxTW1 and the distances between potential salt bridges. From Fig. 1B it seems that the third conserved residue is at an appropriate distance to interact with the substrate in the -1 subsite. This is in agreement with observations of Pozzo and coworkers (25) who proposed an undetermined catalytic role for this residue. A more detailed discussion on what is known about this third residue has been elaborated above (Reviewer #2, comment 2) and a brief commentary has been also included in the manuscript (pages 4-5, lines 97-101).

The reactions to measure product yield are often at a pH close to the pK_a as proposed earlier to be important. That would mean that despite remaining activity for substances with a rather high pK_a would probably rarely lead to full conversion?

The question is very interesting, but it is not easy to give a general answer. From our results, it seems clear that for acceptors with pK_a > 9 there is a dramatic drop in glycosylation yield, probably because the pK_a is too high to allow the deprotonation in the active site, even with the assistance theoretically provided by the third acid residue. However, within the range from 2 to 9 the glycosylation yield appears to be more related to the capacity of the enzyme for hydrolyzing the newly formed glycosidic bond than to the pK_a of the acceptor (page 13, lines 320-322). Thus, in the case of 1*H*-1,2,3-triazolo[4,5-*b*]pyridine ~99% conversion was achieved despite having a pK_a as high as 8.70 (Fig. 4A). This occurs probably because deprotonation in the active, though unfavorable at the reaction pH, is not so severely impaired as it is for compounds with pK_a > 9 and, very importantly, the corresponding xylosyl derivative once formed, cannot be enzymatically hydrolyzed. Thus, the formation of the new glycosidic N-C bond becomes virtually irreversible and forces the continuous consumption of the acceptor. Even so, the proposal of a pH close to pK_a as optimal is still valid as it can be confirmed by the very same example. Thus, conversion was slower at pH 3.0 and it was stagnant at ~20% with pH 2.2.

Conversely, vanillin, which may seem an acceptor with a more convenient pK_a (7.40) reached a maximum conversion between 60-70% (Fig. 4A), probably because the new glycoconjugate is still susceptible of sluggish hydrolysis by the mutant. Nevertheless, in this case, the reaction pH also led to the expected results and maximum yields were attained at pH 5.5.

Line 246: deravatives \diamond derivatives Corrected (page 14, line 338).

Line 261: Why was vanillin used for arabinofuranose. Vanillin wasn't the best acceptor for xylose either?

As the reviewer indicates, vanillin was not the acceptor for which maximum conversion values were achieved using *p*NPX as donor, but it was selected as acceptor of L-arabinopyranose and L-arabinofuranose based on a number of technical and biological reasons.

From the operational perspective, i) the acceptor is an inexpensive and simple phenolic compound with a single group susceptible of being glycosylated; ii) no self-degradation of the vanillin aglycon was observed in the reaction conditions after the longest time of incubation; iii) the xylosyl derivative achieved relatively high (>60%) conversion values; iv) its phenolic nature permits rapid visualization under UV light in the TLC analyses; v) the expected arabinosyl derivatives were, theoretically, still susceptible of being hydrolyzed by rBxTW1-E495. Consequently, this acceptor allowed a very straight-forward glycosylation assay, with, potentially, one single glycoside as product. Simultaneously, it offered the opportunity of studying not only the formation but also the loss of the arabinopyranosyl derivative by the enzymatic action, which would not be possible if an acceptor forming a non-hydrolysable bond had been selected (e.g. thiols or triazoles).

From the biological point of view, vanillin possesses undoubted significance, especially in the food industry but also as a bioactive compound with pharmacological potential. In a recent study, we reported the enzymatic synthesis of a glucosyl derivative of vanillin showing both enhanced antiproliferative capacity and biosafety (13). Therefore, it was considered that vanillin was a

better choice to assay the formation of arabinosyl derivatives than other acceptors with potentially higher conversion yields, but less studied biological behavior, such as thiols or triazoles.

We thank the reviewer's comment on this matter since we realize that these reasons are not obvious if not stated. A brief commentary has been included in the manuscript (page 15, line 362-365) to clarify the motives for selecting vanillin.

How good is arabinofuranose in the wildtype? How much would a lower K_m and therefore the binding and formation of the glycosylenzyme intermediate influence the reaction? Is the relative glycoside formation rate actually the same if the intermediate is actually formed?

Neither arabinofuranose nor arabinopyranose are preferred substrates for the wild-type enzyme in terms of K_m , whereas k_{cat} is similar between *p*NPAf and *p*NPX (24).

Regarding the second question, a kinetic study was performed for rBxTW1-E495A using *p*NPAP (Supplementary, page 6, lines 130-134) and it is interesting to visualize the differences between wild-type and mutant for *p*NPX and *p*NPAP kinetically characterized using acetate buffer.

Enzyme	p NPX		p NPAP		Difference
	K_m (mM)	k_{cat} (s ⁻¹)	K_m (mM)	k_{cat} (s ⁻¹)	
rBxTW1	0.20	69.3	1.6	28	
rBxTW1-E495A	0.67	2.41	5.0	0.097	
	3.3-fold	28.8-fold	3.1-fold	288.7-fold	

The data show that the effect of the mutation on the K_m is virtually the same, but reaction rate drops one additional order of magnitude in the case of *p*NPAP. Considering that both donors contain *p*NP as the leaving group, in our opinion, the formation of the intermediate must occur in a similar manner, but the arabinopyranosyl-enzyme intermediate is affecting the suitable positioning of the acceptor (in this case, acetate from the buffer) to carry out a nucleophilic attack. Even though the reviewer's question is about arabinofuranose instead of arabinopyranose, we anticipate even a more dramatic decrease in k_{cat} . However, an additional deleterious contribution to the K_m cannot be ruled out either at this point, but the enzyme quantities required to determine proper kinetics are impractically high. Based on this reasoning, it seems the main impairment for glycoside formation falls on the deglycosylation of the arabinosyl-enzyme intermediate, and therefore a lower K_m value would probably not suppose a significant enhancement in the overall glycosylation yields, especially taking into account that the initial concentration of donor in those assays was as high as 20 mM.

Figures

Figure 3: some of the figures activities don't start at 0 activity, which is auto hydrolysis? Is the different starting activity solely due to the different reaction pH?

The activity of rBxTW1-E495A in the absence of an acceptor is low but higher than 0. Table S7 displays a k_{cat} value of 0.313 s⁻¹ for the alanine mutant using sodium citrate buffer (considered as another non-interfering buffer due to its negative formal charge at pH 5). This value is much lower than the one attained for the wild-type enzyme (69.3 s⁻¹) but still measurable. This is not a

phenomenon exclusive for the biocatalyst selected in the present work. Mutants in the acid/base catalyst have been reported to show a dramatic decrease of reaction velocity in comparison to their wild-type counterparts but a low activity remains even in the absence of nucleophiles when a nitrophenyl-sugar substrate is used (68). In terms of reaction mechanism, this residual hydrolytic activity implies that water is still able to perform a nucleophilic attack on the glycosyl-enzyme intermediate, but at a much lower rate in comparison with a proper acceptor.

Autohydrolysis of *p*NPX is almost negligible but it was also determined spectrophotometrically, subtracting the absorbance of control replicates prepared in the same conditions, but without enzyme, for every measurement. Consequently, autohydrolysis does not contribute to the activities displayed in the figure. This technical comment has been added to the Supplementary Information (page 3, lines 75-77).

The small variations in the starting activity of rBxTW1-E495A may be attributed to the different pH as the reviewer suggested, showing higher values at pH 2.2 and 3 than pH 5.5, in good agreement with the preference of the wild-type enzyme for the acidic pH values (24). It should be noted that the differences on the starting activity of the wild-type β -xylosidase are much more extreme. The phenomenon is not surprising, not only due to its strong preference for low pH values (24), but also because of the use of ethanol and methanol as co-solvents for some acceptors, since alkan-1-ols are transxylosylation acceptors for rBxTW1 (72).

Figure 4: Similar issue. Here most certainly without acceptor the yield should be zero.? If we understand the question correctly, the reviewer refers to a figure plotting acceptor concentration and yield. In that case, as the comment suggests, the yield should be zero in the absence of acceptor. However, in this case neither figure 4A nor 4B show the combination of those two parameters.

Figure 4B displays specific activity (y-axis) vs acceptor concentration (x-axis). The values for the activities without acceptor should be interpreted as already commented in the previous answer. Those values correspond to the residual but still measurable hydrolytic rates against nitrophenyl-sugar substrates of glycosidase mutants in the acid/base catalyst.

Conversely, figure 4A displays the conversion of 20 mM acceptor (y-axis) along time (x-axis). In this case, the figure shows that at time zero, the yield is zero, too.

We have added a dashed line to make the distinction between the subfigures clearer. Also, we apologize in case we have misinterpreted the meaning of this comment.

Supplementary Material Supplementary Results

It is mentioned that the third residue might be involved in the deprotonation of the incoming receptor. Is that residue close enough to the +1 subsite or the bond to be formed/ cleaved to be of relevance. It looks too far away?

We agree on the fact that the third residue may seem far from the bond to be cleaved. Our measurements using the structure of the highly homologous GH3 β -glucosidase from *Hypocrea*

jecorina and the associated β -D-glucose ligand (pdb: 3ZYZ) indicate that the residue and the anomeric hydroxyl group are around 6.6 Å apart. However, the position of the acceptor in the active site is not clear at all, which could significantly reduce this distance. Moreover, removing the catalytic acid/base residue greatly expands the number and nature of potential acceptors for rBxTW1-E495A. This acquired promiscuity suggests that the mutation of the catalytic acid/base residue might loosen the structural constraints for the entrance of acceptors. In this new context, it is hard to predict how the acceptor is positioned in the active site and its distance to the third acidic residue. We would like to remark that the role attributed to this residue is still a hypothesis. However, the collected data strongly suggest that the acceptor must enter in the neutral form and then it is deprotonated in the active site, and among those residues close enough to the catalytic residues, E91 seems to be the most adequate candidate to participate in such deprotonation.

It would be interesting to discuss a bit the +1 subsite, which seems to show remarkable plasticity to accept so many different acceptors.

This reviewer's question is suggestive and, although there is a lack of information on this regard for GH3 glycosidases, some common trends can be drawn based on reports from other thioglycoligases. Thus, Tegl and coworkers (73) observed that a thioglycoligase mutant from a GH20 hexosaminidase (SpHex) displayed broad acceptor specificity, a phenomenon also observed for the thioglycoligase Abg E171A (74, 75). This greater promiscuity was associated to an increase in space within the subsite +1 caused by the mutation of the catalytic acid/base residue. In the case of SpHex, the authors reported a net expansion of 3.5 Å of the binding pocket upon the E314A mutation, suggesting that this remarkably loosens the constraints of steric hindrance (73). Focusing on the model of rBxTW1, assuming the same architecture of the active site, the E495A mutation would widen the binding pocket by 3.3 Å, therefore the same rationale may apply to explain the enhanced acceptor tolerance of the mutant. In addition, the model of rBxTW1 also indicates that the subsite +1 is considerably shallow, being limited by a few residues (Q15, S16, Q17, P18, L25 and R99), which may ease the approach of the acceptor to the active site. A brief commentary summarizing this discussion has been added to the manuscript (page 7 line 164-172).

Line 440. That would mean acetic acid is an acceptor for the transglycosylation reaction. Isn't that just corroborating the results in the main part that you need MES or Glycine to have no influence of the buffer.

As the reviewer suggests the results from acetic acid indicated that it was acting as a glycosylation acceptor, corroborating the need of using buffering compounds with formal charge to avoid interference with the reaction. However, we consider this assay interesting since it was the first finding suggesting there was a relation between the state of the acceptor (neutral or charged) and the development of the glycosylation reaction catalyzed by rBxTW1-E495A. Additional information has been incorporated into the manuscript, before the main results, summarizing the preliminary characterization of the mutant, including the assay mentioned by reviewer's comment (page 6, lines 133-146). The additional text refers to the corresponding supplementary results with the purpose of improving the logical flow of the manuscript. Moreover, a mention of this result has been added to the discussion of results for MES and glycine in the manuscript (page 9, lines 202-205).

Line445: same underlying mechanism. What is meant with that.

It refers to the fact that despite the removal of the catalytic acid/base causes a dramatic loss of k_{cat} , trapping the biocatalyst in the glycosyl-enzyme intermediate also leads to the decrease of K_m . Consequently, the perturbation of the k_{cat}/K_m is highly dependent on the selected substrate and it is not surprising that the different mutants in both buffers share a similar k_{cat}/K_m value.

However, after reviewing the document we understand that this does not add relevant information for the discussion of rBxTW1-E495A and may be confusing as stated, therefore we have removed the sentence (Supplementary, page 25, line 431).

Line 456 to 460: I presume that was the first test, if the enzyme is able to act as thioglycoligase. It seems for the logical flow better to put that as a statement in the main section.

As in the previous comment about acetic acid, we have referred to this result in the additional information incorporated into the manuscript (page 6, lines 133-146; page 7, lines 155-157) to clarify the rationale behind the assays performed.

Supp Results 2.2 seems redundant, as this is discussed in the main section and is also supported by the effects with glutamate, described in the main section.

Following the reviewer's suggestion, the discussion of Supplementary Results 2.2 has been removed (Supplementary, pages 27-28, lines 455-463). The figure has been maintained (page 28, Fig. S4), in case the reader was interested in corroborating that increasing concentrations of MES have no effect on the mutant's activity.

Supp Results 2.4 Again explains in more detail what is going on with other sugars and indeed propose the theory that the inherent lower affinity towards different sugars might be responsible for the different results.

Attending to the reviewer's comment the full assay and the corresponding discussion has been moved from the Supplementary to the main manuscript (pages 15-16, lines 359-382; Fig. 6).

That reinforces the perceived disconnection between the main and the supplementary part of the manuscript. Some of the results in the main section could go into the supplementary and supplementary results should go into the main part to improve the logic flow and understandability of the manuscript.

Following the reviewer's indications, the manuscript has been modified in order to state the logic flow of the assays carried out, and to connect the preliminary experiments displayed in the Supplementary results with the main text (page 6, lines 133-146; page 7, lines 155-157). In addition, in response to the referees' suggestions, Fig. S1 has been moved to the manuscript as Figs. 1B and 1C, together with all the details on the synthesis of arabinosyl derivatives (pages 15-16, lines 359-382; Fig. 6). We hope that the corrected versions of the Supplementary and the main text convey more clearly the rationale of the experimental design and the significance of these findings.

REFERENCES

1. M. Almendros *et al.*, Exploring the synthetic potency of the first furanothioglycoligase through original remote activation. *Org Biomol Chem* **9**, 8371-8378 (2011).

2. S. Torpenholt *et al.*, Activity of three β -1,4-galactanases on small chromogenic substrates. *Carbohydr Res* **346**, 2028-2033 (2011).
3. J. Ati, P. Lafite, R. Daniellou, Enzymatic synthesis of glycosides: from natural *O*- and *N*-glycosides to rare *C*- and *S*-glycosides. *Beilstein J Org Chem* **13**, 1857-1865 (2017).
4. L.-Z. Lin, J. M. Harnly, A screening method for the identification of glycosylated flavonoids and other phenolic compounds using a standard analytical approach for all plant materials. *J Agric Food Chem* **55**, 1084-1096 (2007).
5. B. Gullón, T. A. Lú-Chau, M. T. Moreira, J. M. Lema, G. Eibes, Rutin: A review on extraction, identification and purification methods, biological activities and approaches to enhance its bioavailability. *Trends in Food Science & Technology* **67**, 220-235 (2017).
6. G. Ren *et al.*, Synthesis of flavonol 3-*O*-glycoside by UGT78D1. *Glycoconj J* **29**, 425-432 (2012).
7. D. Treutter, Significance of flavonoids in plant resistance: A review. *Environ Chem Lett* **4**, 147 (2006).
8. A. N. Panche, A. D. Diwan, S. R. Chandra, Flavonoids: An overview. *J Nutr Sci* **5**, (2016).
9. W. Wang *et al.*, The biological activities, chemical stability, metabolism and delivery systems of quercetin: A review. *Trends in Food Science & Technology* **56**, 21-38 (2016).
10. F. Yin *et al.*, Silibinin: A novel inhibitor of A β aggregation. *Neurochem Int* **58**, 399-403 (2011).
11. G. Deep, R. Agarwal, Antimetastatic efficacy of silibinin: molecular mechanisms and therapeutic potential against cancer. *Cancer Metastasis Rev* **29**, 447-463 (2010).
12. L. Chakrawarti, R. Agrawal, S. Dang, S. Gupta, R. Gabrani, Therapeutic effects of EGCG: A patent review. *Expert Opin Ther Pat* **26**, 907-916 (2016).
13. J. A. Méndez-Lítez *et al.*, Transglycosylation products generated by *Talaromyces amestolkiae* GH3 β -glucosidases: Effect of hydroxytyrosol, vanillin and its glucosides on breast cancer cells. *Microb Cell Fact* **18**, 12 (2019).
14. M. Nieto-Domínguez *et al.*, Enzymatic synthesis of a novel neuroprotective hydroxytyrosyl glycoside. *J Agric Food Chem* **65**, 10526-10533 (2017).
15. P. Torres *et al.*, Enzymatic synthesis of α -glucosides of resveratrol with surfactant activity. *Adv Synth Catal* **353**, 1077-1086 (2011).
16. Y.-H. Moon *et al.*, Synthesis, structure analyses, and characterization of novel rpgalocatechin gallate (EGCG) glycosides using the glucansucrase from *Leuconostoc mesenteroides* B-1299CB. *J Agric Food Chem* **54**, 1230-1237 (2006).
17. P. Kosina *et al.*, Antioxidant properties of silybin glycosides. *Phytother Res* **16**, 33-39 (2002).
18. J. Y. Seo *et al.*, Quercetin 3-*O*-xyloside ameliorates acute pancreatitis in vitro via the reduction of ER stress and enhancement of apoptosis. *Phytomedicine* **55**, 40-49 (2019).
19. E. S. Scarpa, E. Antonini, F. Palma, M. Mari, P. Ninfali, Antiproliferative activity of vitexin-2-*O*-xyloside and avenanthramides on CaCo-2 and HepG2 cancer cells occurs through apoptosis induction and reduction of pro-survival mechanisms. *Eur J Nutr* **57**, 1381-1395 (2017).
20. J. H. Yoon, S. B. Hong, S. J. Ko, S. H. Kim, Detection of extracellular enzyme activity in *Penicillium* using chromogenic media. *Mycobiology* **35**, 166-169 (2007).
21. A. Fiorentino *et al.*, Kaempferol glycosides from *Lobularia maritima* and their potential role in plant interactions. *Chem Biodivers* **6**, 204-217 (2009).
22. J. A. Ozga, A. Saeed, W. Wismer, D. M. Reinecke, Characterization of cyanidin- and quercetin-derived flavonoids and other phenolics in mature saskatoon fruits (*Amelanchier alnifolia* Nutt.). *J Agric Food Chem* **55**, 10414-10424 (2007).
23. M. Gulluce *et al.*, Isolation of some active compounds from *Origanum vulgare* L. ssp. *vulgare* and determination of their genotoxic potentials. *Food Chem* **130**, 248-253 (2012).
24. M. Nieto-Domínguez *et al.*, Enzymatic fine-tuning for 2-(6-hydroxynaphthyl) β -D-xylopyranoside synthesis catalyzed by the recombinant β -xylosidase BxTW1 from *Talaromyces amestolkiae*. *Microb Cell Fact* **15**, 1-15 (2016).

25. T. Pozzo, J. L. Pasten, E. N. Karlsson, D. T. Logan, Structural and functional analyses of β -glucosidase 3B from *Thermotoga neapolitana*: A thermostable three-domain representative of glycoside hydrolase 3. *J Mol Biol* **397**, 724-739 (2010).
26. Y.-K. Li, J. Chir, S. Tanaka, F.-Y. Chen, Identification of the general acid/base catalyst of a family 3 β -Glucosidase from *Flavobacterium meningosepticum*. *Biochemistry* **41**, 2751-2759 (2002).
27. M. W. Zmudka, J. B. Thoden, H. M. Holden, The structure of DesR from *Streptomyces venezuelae*, a β -glucosidase involved in macrolide activation. *Protein Sci* **22**, 883-892 (2013).
28. J. N. Varghese, M. Hrmova, G. B. Fincher, Three-dimensional structure of a barley β -D-glucan exohydrolase, a family 3 glycosyl hydrolase. *Structure* **7**, 179-190 (1999).
29. T. Matsuzawa, M. Watanabe, Y. Nakamichi, Z. Fujimoto, K. Yaoi, Crystal structure and substrate recognition mechanism of *Aspergillus oryzae* isoprimeverose-producing enzyme. *J Struct Biol* **205**, 84-90 (2019).
30. S. P. Schröder *et al.*, Dynamic and functional profiling of xylan-degrading enzymes in *Aspergillus* secretomes using activity-based probes. *ACS Cent Sci* **5**, 12 (2019).
31. Y. Nakatani, S. M. Cutfield, N. P. Cowieson, J. F. Cutfield, Structure and activity of exo-1,3/1,4- β -glucanase from marine bacterium *Pseudoalteromonas* sp. BB1 showing a novel C-terminal domain. *FEBS J* **279**, 464-478 (2012).
32. Y.-W. Kim, R. Zhang, H. Chen, S. G. Withers, *O*-Glycoligases, a new category of glycoside bond-forming mutant glycosidases, catalyse facile syntheses of isoprimeverosides. *Chem Commun* **46**, 8725-8727 (2010).
33. C. Li, J.-H. Kim, Y.-W. Kim, α -Thioglycoligase-based synthesis of *O*-aryl α -glycosides as chromogenic substrates for α -glycosidases. *J Mol Catal B Enzym* **87**, 24-29 (2013).
34. H.-J. Ahn *et al.*, Enzymatic synthesis of 3-*O*- α -maltosyl-L-ascorbate using an engineered cyclodextrin glucanotransferase. *Food Chem* **169**, 366-371 (2015).
35. C. Li, H. J. Ahn, J. H. Kim, Y. W. Kim, Transglycosylation of engineered cyclodextrin glucanotransferases as *O*-glycoligases. *Carbohydr Polym* **99**, 39-46 (2014).
36. S. Karkehabadi *et al.*, Biochemical characterization and crystal structures of a fungal family 3 β -glucosidase, Cel3A from *Hypocrea jecorina*. *J Biol Chem* **289**, 31624-31637 (2014).
37. M. N. Namchuk, J. D. McCarter, A. Becalski, T. Andrews, S. G. Withers, The role of sugar substituents in glycoside hydrolysis. *J Am Chem Soc* **122**, 1270-1277 (2000).
38. C. Bohlin *et al.*, A comparative study of hydrolysis and transglycosylation activities of fungal β -glucosidases. *Appl Microbiol Biotechnol* **97**, 159-169 (2013).
39. E. V. Eneyskaya *et al.*, Biochemical and kinetic analysis of the GH3 family β -xylosidase from *Aspergillus awamori* X-100. *Arch Biochem Biophys* **457**, 225-234 (2007).
40. P. Turner, D. Svensson, P. Adlercreutz, E. N. Karlsson, A novel variant of *Thermotoga neapolitana* β -glucosidase B is an efficient catalyst for the synthesis of alkyl glucosides by transglycosylation. *J Biotechnol* **130**, 67-74 (2007).
41. K. Suzuki *et al.*, Crystal structures of glycoside hydrolase family 3 β -glucosidase 1 from *Aspergillus aculeatus*. *Biochem J* **452**, 211-221 (2013).
42. M.-R. Kao *et al.*, *Chaetomella raphigera* β -glucosidase D2-BGL has intriguing structural features and a high substrate affinity that renders it an efficient cellulase supplement for lignocellulosic biomass hydrolysis. *Biotechnol Biofuels* **12**, 1-18 (2019).
43. J. Agirre *et al.*, Three-dimensional structures of two heavily *N*-glycosylated *Aspergillus* sp. family GH3 β -d-glucosidases. *Acta Crystallographica Section D: Structural Biology* **72**, 254-265 (2016).
44. S. Karkehabadi *et al.*, Biochemical characterization and crystal structures of a fungal family 3 β -glucosidase, Cel3A from *Hypocrea jecorina*. *J Biol Chem* **289**, 31624-31637 (2014).
45. K. V. Mahasenan *et al.*, Catalytic cycle of glycoside hydrolase BglX from *Pseudomonas aeruginosa* and its implications for biofilm formation. *ACS Chem Biol* **15**, 189-196 (2020).
46. M. Ramírez-Escudero *et al.*, Structural and functional characterization of a ruminal β -glycosidase defines a novel subfamily of glycoside hydrolase family 3 with permuted domain topology. *J Biol Chem* **291**, 24200-24214 (2016).

47. S. Yan *et al.*, Functional and structural characterization of a β -glucosidase involved in saponin metabolism from intestinal bacteria. *Biochem Biophys Res Commun* **496**, 1349-1356 (2018).
48. H. F. Seidle, K. McKenzie, I. Marten, O. Shoseyov, R. E. Huber, Trp²⁶² is a key residue for the hydrolytic and transglucosidic reactivity of the *Aspergillus niger* family 3 β -glucosidase: Substitution results in enzymes with mainly transglucosidic activity. *Arch Biochem Biophys* **444**, 66-75 (2005).
49. A. M. Gumel, M. S. M. Annuar, T. Heidelberg, Y. Chisti, Lipase mediated synthesis of sugar fatty acid esters. *Process Biochem* **46**, 2079-2090 (2011).
50. K. Ren, B. P. Lamsal, Synthesis of some glucose-fatty acid esters by lipase from *Candida antarctica* and their emulsion functions. *Food Chem* **214**, 556-563 (2017).
51. J. O. Rich, B. A. Bedell, J. S. Dordick, Controlling enzyme-catalyzed regioselectivity in sugar ester synthesis. *Biotechnol Bioeng* **45**, 426-434 (1995).
52. E. Abdulmalek, N. F. Hamidon, M. B. Abdul Rahman, Optimization and characterization of lipase catalysed synthesis of xylose caproate ester in organic solvents. *J Mol Catal B Enzym* **132**, 1-4 (2016).
53. M. R. Couri *et al.*, Microwave-assisted efficient preparation of novel carbohydrate tetrazole derivatives. *Carbohydr Res* **342**, 1096-1100 (2007).
54. S. B. Ferreira *et al.*, Synthesis, biological activity, and molecular modeling studies of 1*H*-1,2,3-triazole derivatives of carbohydrates as α -glucosidases inhibitors. *J Med Chem* **53**, 2364-2375 (2010).
55. R. R. Kale, V. Prasad, D. Kushwaha, V. K. Tiwari, Benzotriazole-mediated facile synthesis of novel glycosyl tetrazole. *J Carbohydr Chem* **31**, 130-142 (2012).
56. K. Slámová *et al.*, Synthesis and biological activity of glycosyl-1*H*-1,2,3-triazoles. *Bioorg Med Chem Lett* **20**, 4263-4265 (2010).
57. B. L. Wilkinson, H. Long, E. Sim, A. J. Fairbanks, Synthesis of arabino glycosyl triazoles as potential inhibitors of mycobacterial cell wall biosynthesis. *Bioorg Med Chem Lett* **18**, 6265-6267 (2008).
58. S. André, K. E. Kövér, H.-J. Gabius, L. Szilágyi, Thio- and selenoglycosides as ligands for biomedically relevant lectins: Valency-activity correlations for benzene-based dithiogalactoside clusters and first assessment for (di)selenodigalactosides. *Bioorg Med Chem Lett* **25**, 931-935 (2015).
59. B. Ayers *et al.*, Stereoselective synthesis of β -arabino glycosyl sulfones as potential inhibitors of mycobacterial cell wall biosynthesis. *Carbohydr Res* **344**, 739-746 (2009).
60. Y.-W. Kim *et al.*, Thioglycoligase-based assembly of thiodisaccharides: Screening as β -galactosidase inhibitors. *ChemBioChem* **8**, 1495-1499 (2007).
61. S. Mehta, J. S. Andrews, B. Svensson, B. M. Pinto, Synthesis and enzymic activity of novel glycosidase inhibitors containing sulfur and selenium. *J Am Chem Soc* **117**, 9783-9790 (1995).
62. F. Castaneda, A. Burse, W. Boland, R. K. Kinne, Thioglycosides as inhibitors of hSGLT1 and hSGLT2: Potential therapeutic agents for the control of hyperglycemia in diabetes. *Int J Med Sci* **4**, 131 (2007).
63. J. Rodrigue *et al.*, Aromatic thioglycoside inhibitors against the virulence factor LecA from *Pseudomonas aeruginosa*. *Org Biomol Chem* **11**, 6906-6918 (2013).
64. V. Puchart, Glycoside phosphorylases: Structure, catalytic properties and biotechnological potential. *Biotechnol Adv* **33**, 261-276 (2015).
65. S. S. Macdonald, M. Blaukopf, S. G. Withers, *N*-acetylglucosaminidases from CAZy family GH3 are really glycoside phosphorylases, thereby explaining their use of histidine as an acid/base catalyst in place of glutamic acid. *J Biol Chem* **290**, 4887-4895 (2015).
66. S. S. Macdonald *et al.*, Structural and mechanistic analysis of a β -glycoside phosphorylase identified by screening a metagenomic library. *J Biol Chem* **293**, 3451-3467 (2018).
67. M. Jahn, S. G. Withers, New approaches to enzymatic oligosaccharide synthesis: Glycosynthases and thioglycoligases. *Biocatal Biotransformation* **21**, 159-166 (2003).

68. Q. Wang, D. Trimbur, R. Graham, R. A. J. Warren, S. G. Withers, Identification of the acid/base catalyst in *Agrobacterium faecalis* b-glucosidase by kinetic analysis of mutants. *Biochemistry* **34**, 14554-14562 (1995).
69. I. Z. Breen *et al.*, in *eLS*. (John Wiley & Sons, Chichester, 2018).
70. F. H. M. Souza, R. F. Inocentes, R. J. Ward, J. A. Jorge, R. P. M. Furriel, Glucose and xylose stimulation of a β -glucosidase from the thermophilic fungus *Hemicola insolens*: A kinetic and biophysical study. *J Mol Catal B Enzym* **94**, 119-128 (2013).
71. W. Zwerschke *et al.*, Allosteric activation of acid α -glucosidase by the human papillomavirus E7 protein. *J Biol Chem* **275**, 9534-9541 (2000).
72. M. Nieto-Domínguez *et al.*, Novel pH-stable glycoside hydrolase family 3 b-xylosidase from *Talaromyces amestolkiae*: An enzyme displaying regioselective transxylosylation. *Appl Environ Microbiol* **81**, 6380-6392 (2015).
73. G. Tegl *et al.*, Facile formation of β -thioGlcNAc linkages to thiol-containing sugars, peptides, and proteins using a mutant GH20 hexosaminidase. *Angew Chem Int Ed Engl* **58**, 1632-1637 (2019).
74. M. Jahn, J. Marles, R. A. J. Warren, S. G. Withers, Thioglycoligases: Mutant glycosidases for thioglycoside synthesis. *Angew Chem Int Ed Engl* **42**, 352-354 (2003).
75. R. V. Stick, K. A. Stubbs, From glycoside hydrolases to thioglycoligases: The synthesis of thioglycosides. *Tetrahedron: Asymmetry* **16**, 321-335 (2005).

REVIEWERS' COMMENTS:

Reviewer #1 (Remarks to the Author):

The authors have fully and greatly replied to all comments. The current manuscript deserves to be reported now.

Reviewer #2 (Remarks to the Author):

The authors have been answered all my comments properly.

Reviewer #3 (Remarks to the Author):

The manuscript., titled "Thioglycoligase derived from fungal GH3 β -xylosidase reveals outstanding acceptor tolerance: From thio- to multi-glycoligases" by Nieto-Dominguez et al. is a resubmission of a revision. Briefly, the manuscript describes the generation of a thioglycoligase based on a GH3 from the *Talaromyces amestolkiae*. The enzyme proves to have an outstanding acceptor tolerance and shows robust activity, accepting acceptors with a wide range of pKa values. The enzyme is able to generate N-, O-, S, Se- and other types of glycoside, making it a promising candidate for the large scale production of a wide range of glycosides.

The authors revised the manuscript carefully and substantially throughout. It reads now much better and brings everything together in a really good story. Overall, the manuscript is close to ready for publication. A few minor changes will help to improve clarity in parts of the manuscript as outlined below.

Abstract

For bonding \diamond to connect

Limited just to \diamond limited to strong nucleophilic acceptors

Glycoconjugates portfolio \diamond glycoconjugate portfolio

Introduction:

Line 44 to 47: That sentence is very long and it would be better to make two out of it.

Line 63 ...is known as double

Displacement mechanism

Line 66: released aglycon \diamond leaving group

I would recommend leaving group, because it can be a aglycon of a glycan, depending on the substrate

Line 68: phase \diamond step

Line 72: a hemiacetal is only formed if the nucleophile is a water molecule. With any other nucleophile an acetal might be formed.

Line 87 is still reduced \diamond is still limited

Line 125 ease research \diamond facilitate research

Line 168 There should be a figure (supplementary of the acceptorsite) with surface representation to show the shallowness and a residues mention. Without this it is hard to comprehend or visualise by itself what is meant and how these residues define the acceptor subsite.

Line 172 without strong nucleophilicity of the acceptor

Line 182 It might be true that at suboptimal pH-values the formation of the glycosylenzyme intermediate is impaired, but it is hard to decouple it from the perceived instability of the enzyme.

Line 185 electrostatic repulsion? Is the active site that strong negatively charged besides the catalytic acid and base to support this claim.

Line 187 Is that plausible? Are there any data to suggest that the formed products would be unstable? Has anyone tried to synthesized them chemically and check their stability?

Line 188 little or no

Line 197 to act as

Line 198 why was expected that increasing MES concentrations would boost the activity?

It is interesting to note from the experiment with glutamic acid that the amino function with a positive charge seem to be solely responsible for decrease activity, despite the fact that the amino group of some amino acids should be in a suitable range to allow for some product formation. The alpha Carboxy function seems more amenable to break the glycosylenzyme. Can that be rationalised?

Line 204-205: I don't understand the conclusion with the citric acid

Line 251- 252 How does that sentence fit here? Also taken into account the explanations regarding the yield of vanillin xyloside in the rebuttal.

Below the argument Is not clear as stated in the rebuttal one can get 99% conversion with a tetrazole having a pKa higher than vanillin

Line 274 in a nonbuffered environment

Line 300 The data itself are not difficult to interpret, but I think what the authors want to elude to, is that there are not much known about the usefulness of these compounds

Line 333 to estimate the most suitable pH

Line 385 aroused \diamond sparked

Supplementary Material

Line 417 glutamic acid residue

REVIEWERS' COMMENTS:

Reviewer #1 (Remarks to the Author):

The authors have fully and greatly replied to all comments. The current manuscript deserves to be reported now.

We are really grateful for reviewer's kind assessment of our work.

Reviewer #2 (Remarks to the Author):

The authors have been answered all my comments properly.

We thank reviewer for the favorable opinion on our work.

Reviewer #3 (Remarks to the Author):

The manuscript, titled "Thioglycoligase derived from fungal GH3 β -xylosidase reveals outstanding acceptor tolerance: From thio- to multi-glycoligases" by Nieto-Dominguez et al. is a resubmission of a revision. Briefly, the manuscript describes the generation of a thioglycoligase based on a GH3 from the *Talaromyces amestolkiae*. The enzyme proves to have an outstanding acceptor tolerance and shows robust activity, accepting acceptors with a wide range of pKa values. The enzyme is able to generate N-, O-, S, Se- and other types of glycoside, making it a promising candidate for the large scale production of a wide range of glycosides. The authors revised the manuscript carefully and substantially throughout. It reads now much better and brings everything together in a really good story. Overall, the manuscript is close to ready for publication. A few minor changes will help to improve clarity in parts of the manuscript as outlined below.

We sincerely appreciate reviewer's opinion on the improvement of the new version of the manuscript. Detailed responses to the suggested corrections and comments are given below.

Abstract

For bonding \square to connect. Corrected (page 1, line 23).

Limited just to \square limited to strong nucleophilic acceptors. Corrected (page 1, lines 24-25).

Glycoconjugates portfolio \square glycoconjugate portfolio. Corrected (page 2, lines 32-33).

Introduction:

Line 44 to 47: That sentence is very long and it would be better to make two out of it. The sentence has been split (page 2, line 45-49).

Line 63 ...is known as double Displacement mechanism. Corrected (page 3, lines 64-65).

Line 66: released aglycon \square leaving group

I would recommend leaving group, because it can be a aglycon of a glycan, depending on the substrate. We agree on the fact that leaving group fits better for the description of the general mechanism. The change has been done (page 3, line 68).

Line 68: phase \square step. Corrected (page 3, line 70).

Line 72: a hemiacetal is only formed if the nucleophile is a water molecule. With any other nucleophile an acetal might be formed. Hemiacetal has been replaced by product, to avoid the ambiguity (page 3, line 73).

Line 87 is still reduced \square is still limited. Corrected (page 4, line 89).

Line 125 ease research □ facilitate research. Corrected (page 6, line 133).

Line 168 There should be a figure (supplementary of the acceptorsite) with surface representation to show the shallowness and a residues mention. Without this it is hard to comprehend or visualise by itself what is meant and how these residues define the acceptor subsite. The requested figure has been added to the Supplementary Information (Supplementary Figure 8) and cited in the main text (page 7, line 174).

Line 172 without strong nucleophilicity of the acceptor. The sentence has been modified (page 8, line 178-179).

Line 182 It might be true that at suboptimal pH-values the formation of the glycosylenzyme intermediate is impaired, but it is hard to decouple it from the perceived instability of the enzyme. We agree with reviewer's observation. The text has been modified to explicitly taking into account both possibilities, which serves to explain the absence of glycosylation for acceptors with pK_a above 10 (page 8, lines 191-193).

Line 185 electrostatic repulsion? Is the active site that strong negatively charged besides the catalytic acid and base to support this claim.

The question is interesting and, as it will be discussed for the comment marked as *Line 251- 252*, results displayed in Figure 4 strongly suggest the neutral form is the preferred for the enzyme whereas yields decrease, dramatically in the case of benzenesulfonic acid, when the equilibrium is displaced to the anion. Therefore, there are experimental evidence suggesting some kind of charge-related phenomenon, like electrostatic repulsion, negatively affects the accommodation of anions in the active site of rBxTW1-E495A.

However, prior to the experimental results, the possibility of electrostatic repulsion should be taken into account based on some considerations of the field. As far as we know there are no reports on anions acting as transglycosylation acceptors for wild-type retaining glycosidases. This absence is extended to strong nucleophiles such as thiols. The phenomenon has barely received attention, although it has been explained as a consequence of electrostatic repulsion. In this sense, the work of Stick and Stubbs¹ is particularly interesting since the glycosylation of thiol acceptors in neutral form was attempted with no success using a wild-type glycosidase and a glycosynthase (mutation of the catalytic nucleophile). The authors suggested that the thiol might get ionized through interacting with the residues of the active site. According to their hypothesis, the formed thiolate would be repulsed by the general/acid base, being unable to act as an acceptor. The present work proposes that the third acid residue of GH3 glycosidases may act as a surrogate of the catalytic acid/base in thioglycoligase mutants, facilitating the deprotonation of glycosylation acceptors. In a similar way, this residue might be also the one responsible for the electrostatic repulsion of anions in the active site.

It should be noticed that other thioglycoligases have been reported to use the anionic form of the acceptor as the reactive species^{2,3}. This reinforces the singularity of rBxTW1-E495A and supports the proposed role for the third acid residue of GH3 glycosidases.

A citation of Stick and Stubbs' work has been added to the manuscript (page 8, line 195), to provide a source for the mentioned electrostatic repulsion.

Line 187 Is that plausible? Are there any data to suggest that the formed products would be unstable? Has anyone tried to synthesized them chemically and check their stability?

As far as we know, there are no reports on the synthesis of the mentioned potential products. However, the sentence seeks to list the main possibilities explaining the absence of glycosylated products. Despite the fact that the further experimental validation (Fig. 3) discards product instability as a valid cause in the case of sulfonic acid acceptor, in our opinion it is reasonable to preliminarily consider this hypothesis. Our reasoning takes into account that both methanesulfonic and ethanesulfonic acids possess low pK_a values (-1.92 and -1.68 respectively), being the strongest acids among the acceptors tested in the present work. Conjugated bases of strong acids are weak and weak bases are indicative of good leaving groups. Consequently, the potential xyloside would contain a good leaving group, which negatively affects the stability of glycoconjugates.

Line 188 little or no. Corrected (page 8, line 198).

Lien197 to act as. Corrected (page 9, lines 206-207).

Line 198 why was expected that increasing MES concentrations would boost the activity?

The rationale here is the one stated to perform the ligase activity response assays. When no glycoconjugate is observed, the possibility of product instability cannot be dismissed. Increasing the concentration of the potential acceptor and check for a boost of the activity is a rapid way to evaluate this possibility. In this case, it was particularly important to discard the role of MES as an acceptor, even an inefficient one, since buffers unable to participate in the catalyzed reactions were needed to gain insight in the behavior of rBxTW1-E495A. Based on the rationale of the manuscript about the zwitterionic nature of MES, the absence of a boost in the activity was indeed the expected result.

It is interesting to note from the experiment with glutamic acid that the amino function with a positive charge seem to be solely responsible for decrease activity, despite the fact that the amino group of some amino acids should be in a suitable range to allow for some product formation. The alpha Carboxy function seems more amenable to break the glycosylenzyme. Can that be rationalised?

As the reviewer implies, amino groups are considered strong nucleophiles in their unprotonated state. However, amino groups of amino acids, both primary and those belonging to the lateral chain, possess pK_a values above 9. Unlike most of the acceptors tested in the present study, amines are bases, which means that at a pH value below their pK_a they are protonated, positively charged. Therefore, in this case, displacing the equilibrium to the neutral form of the acceptor would require a reaction pH at least above 9, which is a value far from the range of 2.2 to 5.5 in which rBxTW1-E495A is active and stable. On the contrary, the alpha carboxyl group possesses a pK_a value of 2.10, therefore at a pH reaction of 2.2, the amount of the acceptor in neutral form should be sufficient to achieve some product synthesis.

However, as reviewer mentioned, the fact that a single product is observed for L-glutamic acid, whereas no product was found for L-glycine and two products were detected for N-acetyl-L-glutamic acid suggests that the positively charged amino group prevents the glycosylation of the alpha carboxyl group. This is consistent with the inability of zwitterions to act as acceptors for rBxTW1-E495A (page 9, lines 202-211).

A sentence has been incorporated into the manuscript to clarify that the mentioned pK_a range from 2 to 9 refers to compounds displaced to the neutral form at a reaction pH below their pK_a value (page 8, lines 185-186).

Line 204-205: I don't understand the conclusion with the citric acid.

The sentence refers to the fact that one of the carboxylic groups of citric acid is displaced to carboxylate form at pH 5.0 ($pK_1 = 3.15$). As discussed above, anions seem to be inadequate acceptors for being glycosylated by rBxTW1-E495A. However, the sentence has been removed to improve clarity. We realize, that the differences between buffers in the kinetic characterization of the mutant are explained in a more straightforward way in the Supplementary Information as a consequence of the disruption of the glycosyl-enzyme intermediate by acetic acid (Supplementary Discussion).

Line 251- 252 How does that sentence fit here? Also taken into account the explanations regarding the yield of vanillin xyloside in the rebuttal.

Below the argument Is not clear as stated in the rebuttal one can get 99% conversion with a tetrazole having a pK_a higher than vanillin.

Reviewer's first question refers to the fact that, in the case of vanillin and 1*H*-1,2,3-triazolo[4,5-*b*]pyridine, higher yields are achieved at pH 5.5 than at lower pH values. This is essentially linked to the relation between pH, pK_a and enzyme activity, which is key for the present work and it is summarized in a series of three requirements that determine the feasibility of the enzymatic glycosylation by rBxTW1-E495A (pages 11-12, lines 274-277). We understand that the comment is related to the third statement, although the following lines also elaborate on the other two to fully contextualize the response.

1) The enzyme should be stable and active, which limits the reaction pH to a range from 2.2 to 5.5.

2) The acceptor should be in an equilibrium displaced to its neutral form. This equilibrium is controlled by the relation between pK_a and reaction pH. The statement is supported by results displayed in Figure 4. Either considering conversion yield or specific activity, values for benzenesulfonic acid, 5-(5-bromo-2-thienyl)-1*H*-tetrazole and 5-trifluoromethyl-2*H*-tetrazole are lower at reaction pH values above pK_a (equilibrium displaced to the anionic form). This is particularly clear for the analysis of glycosylation yield, whereas for the measurements of the specific activity it seems that high concentrations of the acceptor can suppress the differences. The latter phenomenon is not surprising, since a large excess of acceptor, even when it is displaced to the anionic form, also increases the concentration of the neutral compound.

3) The reaction pH should be compatible with the deprotonation of the acceptor in the active site. From the preference for neutral acceptors, it may be assumed that those are the ones suitable for being accommodated in the active site. However, the disruption of the glycosyl-enzyme intermediate, and the formation of a new glycosidic bond with the acceptor, requires the latter to lose a proton from the functional group involved in the reaction. In other words, neutral form is required but not enough. Once in the suitable position, the acceptor should be also able to be deprotonated. According to our hypothesis, the third acid residue might receive this proton from

the acceptor, but reaction pH will facilitate or difficult the process. Results for glycosylation yields of vanillin and 1*H*-1,2,3-triazolo[4,5-*b*]pyridine (Fig. 4A) can be interpreted under the light of the former rationale. Attending to their pK_a values (7.40 and 8.70 respectively), the acceptors are displaced to the neutral form for the three values of pH assayed (2.2, 3.0 and 5.5), although the highest yields are observed for pH 5.5 in both cases. We interpret that deprotonation in the active site is less impaired as the pH value is closer to the pK_a of the acceptor, explaining why lower conversion is achieved at pH 3.0 and pH 2.2.

The manuscript develops a slightly shorter version of the former reasoning (page 11, lines 263-274), although an additional sentence has been incorporated after the lines indicated by the reviewer. The additional text indicates that, despite the lower performance of neutral acceptors at the most acidic pH values may seem paradoxical, those results will be properly discussed in the following paragraph (page 11, lines 260-262).

The reviewer also indicates that the 99% conversion achieved for the 1*H*-1,2,3-triazolo[4,5-*b*]pyridine is surprising, taking into account that its pK_a value is higher than the one associated to vanillin, for which glycosylation yield is more modest. To clarify the response given in the former rebuttal, it should be noticed that both acceptors comply with the three proposed criteria to achieve glycosylation catalyzed by rBxTW1-E495A. i) Reaction is carried out at pH 5.5, ii) acceptors are displaced to the neutral form ($pH < pK_a$). iii) Acceptors are deprotonated in the active site, since the corresponding xylosides are detected. As the reviewer implies, deprotonation may be more favorable for vanillin due to its lower pK_a value. However, since the reaction is feasible for both acceptors, glycosylation yield may be affected by other properties. In this case, the inability of the enzyme to disrupt the formed N-xyloside seems to be a remarkable driving force towards complete conversion. Thus, complete conversion is also achieved for N-hydroxybenzotriazole, thiophenol and 3,5-dibromotriazole, whose corresponding xylosides are not expected to be hydrolysable by rBxTW1-E495A.

As a final remark, we would like to highlight that the relation between pH and pK_a is a measure of the feasibility of the glycosylation reaction, rather than an indicator of the expected yield. Intrinsic differences between acceptors, such as structural features or the nature of the expected glycosidic linkage may affect the final yield. Thus, EGCG and vanillin display close pK_a values, although the latter achieves a considerably higher conversion yield (Fig. 5).

Line 274 in a nonbuffered environment. Corrected (page 12, lines 286-287).

Line 300 The data itself are not difficult to interpret, but I think what the authors want to elude to, is that there are not much known about the usefulness of these compounds. We apologize if the statement seemed elusive. The sentence has been rewritten to clearly establish that little is known about the usefulness of glycosyl derivatives of triazoles and tetrazoles, although almost countless different glycoconjugates may be potentially synthesized due to the huge diversity of these aglycons (page 13, lines 311-313).

Line 333 to estimate the most suitable pH. Corrected (page 14, line 346).

Line 385 aroused \square sparked. Corrected (page 17, line 402).

Supplementary Material

Line 417 glutamic acid residue. Corrected (Supplementary Discussion).

REFERENCES

- 1 Stick, R. V. & Stubbs, K. A. From glycoside hydrolases to thioglycoligases: The synthesis of thioglycosides. *Tetrahedron: Asymmetry* **16**, 321-335, (2005).
- 2 Armstrong, Z., Reitinger, S., Kantner, T. & Withers, S. G. Enzymatic thioxyloside synthesis: Characterization of thioglycoligase variants identified from a site-saturation mutagenesis library of *Bacillus circulans* xylanase. *ChemBioChem* **11**, 533-538, (2010).
- 3 Mullegger, J., Jahn, M., Chen, H. M., Warren, R. A. J. & Withers, S. G. Engineering of a thioglycoligase: randomized mutagenesis of the acid-base residue leads to the identification of improved catalysts. *Protein Eng. Des. Sel.* **18**, 33-40, (2005).